# A mathematical model suggests collectivity and inconstancy enhance the efficiency of neuronal migration in the adult brain

**Daiki Wakita** [iD][1,2], **Yuriko Sobu**[3], **Naoko Kaneko** [iD][3*], **Takeshi Kano** [iD][4*]

**1** Misaki Marine Biological Station, The University of Tokyo, Miura, Kanagawa, Japan, **2** Japan Society for the Promotion of Science, Chiyoda, Tokyo, Japan, **3** Laboratory of Neuronal Regeneration, Graduate School of Brain Science, Doshisha University, Kyotanabe, Kyoto, Japan, **4** School of Systems Information Science, Future University Hakodate, Hakodate, Hokkaido, Japan

* tkano@fun.ac.jp (TK); nkaneko@mail.doshisha.ac.jp (NK)

**Data availability statement:** All relevant data are within the manuscript and its Supporting information files.

## Abstract

Neuronal regeneration in the adult brain, which is restricted compared to that in the embryonic brain, is a long-standing topic in neuroscience and medical research. Based on studies in mammals, a small number of newly generated immature neurons (neuroblasts) migrate toward damaged sites and contribute to functional recovery. During migration, neuroblasts form chain-like collectives and modify the morphology of glial cells (astrocytes), which are the main components of the surrounding environment. However, it remains unclear how neuroblasts form collectives and how efficient migration is achieved through collective formation in a pool of astrocytes. The main difficulty lies in tracking individual neuroblasts within the collective, both in vitro and in vivo, over a period. To address this impasse, we built a mathematical model of the neuroblast-astrocyte system to assess its long-term performance in silico. Our simulations showed that individual neuroblasts gathered into chain-like collectives through occasional contact, astrocyte confinement, and moderate adhesion between the neuroblasts. The forward movement of neuroblasts in an astrocyte-dense environment was accelerated if we assumed a simple interaction: the higher the number of neuroblasts near an astrocyte, the stronger the shrinkage of astrocytic protrusions. Furthermore, temporal changes in neuroblast behavior, as indicated by our observation of living neuroblasts in culture, reinforce the advantages of simulated collectives. A collective of neuroblasts with constant behavior sometimes repeated non-migratory movements, whereas those with inconstant behavior were easily untangled, resulting in a rapid migration. These results highlight the potential for neuroblast collectivity and inconstancy in enhancing neuronal regeneration in the adult brain.

## Author summary

Increasing the regenerative ability of the adult brain is challenging for humans. Only a limited number of newly generated nerve cells (neurons) migrate toward injured regions

**Funding:** This study was supported by JSPS KAKENHI JP21H05104 to TK, JP21H05106 to NK, JP23H01138 to TK, JP23K25835 to TK, JP23K27270 to NK, and JP24K18282 to YS (https://www.jsps.go.jp/english/e-grants/). This study was also supported by the JST FOREST Program JPMJFR2146 to NK (https://www.jst.go.jp/souhatsu/en/index.html), the Naito Foundation to NK (https://www.naito-f.or.jp/en/), and the Mitsubishi Foundation to NK (https://www.mitsubishi-zaidan.jp/en/index.html). The funders had no role in study design, data collection and analysis, decision to publish, or preparation of the manuscript. DW received a salary from JP21H05104.

**Competing interests:** The authors have declared that no competing interests exist.

to participate in the functional regeneration of the adult mammalian brain. During this journey, neurons gather and modify the shape of the surrounding glial cells. Because it is difficult to observe how actual neurons within a group efficiently move in the brain for a long time, we sought to determine the key to rapid migration using a mathematical approach instead of a biological one. Computer simulations showed that, first, neurons form a chain-like group by gently sticking to each other and following the rail-like guide of glial cells. Second, a group of neurons migrates faster than a single one because they can shrink the processes of nearby glial cells more effectively than a single one. Third, a group becomes faster when the behavior of neurons varies over time, even at the same average speed. Our novel concept posits that high regeneration ability in the brain is achieved through the grouped, temporally varying migration of neurons.

## 1. Introduction

Neuronal regeneration in the mammalian brain has long been a challenge for neuroscientists and medical researchers [1,2]. Most neurons are generated during embryonic brain development. However, neural stem cells in the ventricular-subventricular zone (V-SVZ) keep producing new neurons after birth [3,4]. The generated immature neurons (neuroblasts) migrate through a specific route known as the rostral migratory stream (RMS) to the olfactory bulb, where they mature and are integrated into the neuronal circuitry [5,6]. Furthermore, after brain damage such as ischemic stroke, some neuroblasts exhibit migratory behavior toward the site of the lesion and contribute to functional recovery [7–9]. Understanding the regulatory mechanisms of cell migration is crucial to enhancing neuronal regeneration.

During migration, each neuroblast exhibits saltatory movement, characterized by the extension of a long leading process in the direction of travel, followed by somal translocation (Fig 1a and S1 Video) [4]. Neuronal migration has been extensively studied in the context of neocortical morphogenesis, in which excitatory and inhibitory neuron precursors migrate individually from the germinal zones at the ventricular surface toward the developing neocortex via radial and tangential migrations, respectively [10]. In contrast, in adult brains, V-SVZ-derived neuroblasts, the only cell type capable of long-distance migration, migrate collectively, maintaining close contact and forming elongated clusters referred to as "chains" [6,11,12] (Fig 1b). While the RMS begins to develop before birth, the distinct "chain migration" pattern takes approximately three weeks to become fully established [13]. The V-SVZ-derived neuroblasts also exhibit chain migration as they travel toward lesion sites [8]. Although the biological significance of this migration pattern is not fully understood, it has been suggested that collective chain migration provides neuroblasts with an advantage in navigating the challenging and less accessible adult brain environment, characterized by densely packed neuronal fibers and glial cells.

Within these chain-like collectives, neuroblasts form discontinuous adherens junctions with one another, as revealed by electron microscopy [6]. Intercellular adhesion within these chains is regulated by various molecules including N-cadherin and PSA-NCAM [14,15]. Live imaging of migrating neuroblasts in acute brain slices and extracellular matrix-enriched gels has demonstrated that the cells within a chain frequently exchange neighbors during migration [16,17], thus indicating that intercellular adhesion is dynamically remodeled.

Adult-born neuroblasts also influence the surrounding microenvironment to enhance their migration efficiency. They actively modify the morphology of nearby astrocytes [8],

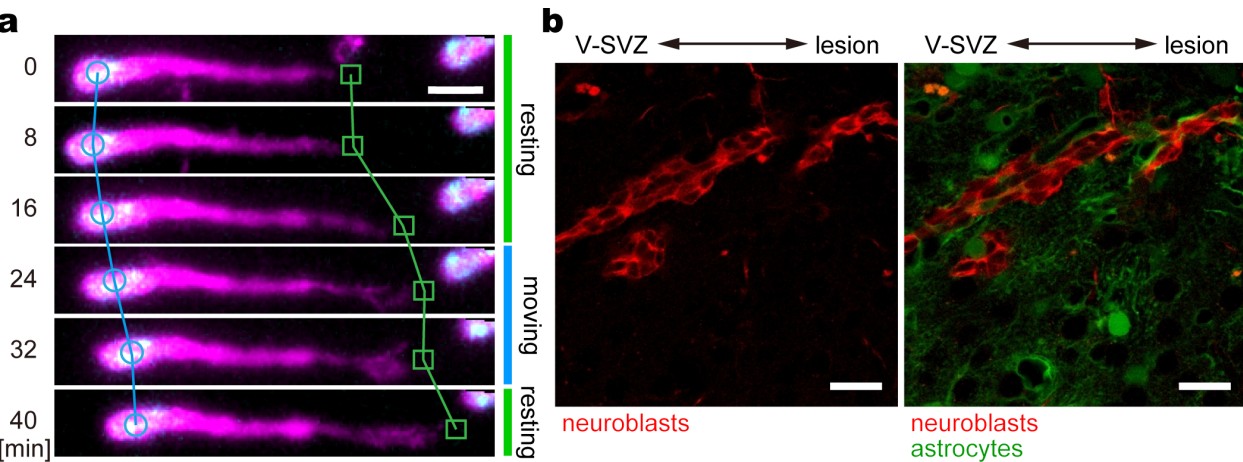

**Fig 1. Living neuroblasts.** (a) Saltatory movement of a single neuroblast. V-SVZ of the Dcx-DsRed mice was cultured in Matrigel. Neuroblasts were identified by DsRed expression (magenta) and the nucleus was labeled by Hoechst 33342 (cyan). The soma was marked with circles, and the tip of the leading process was marked with rectangles. Scale bar = 10 μm. (b) Collective migration of neuroblasts in the mouse brain after stroke. Neuroblasts forming elongated clusters (red) migrate from the V-SVZ through astrocyte (green)-rich tissue toward the lesion. Scale bar = 20 μm. Methods are detailed in S1 Appendix (Sect I).

primary glial cells harboring many processes. Neuroblasts secrete the chemorepellent Slit1, which inhibits process formation in surrounding astrocytes that express its receptor Robo2. This neuroblast-astrocyte interaction is essential for efficient migration both through the RMS under physiological conditions [18] and toward damaged areas following ischemic stroke [8].

When taken together, these findings indicate that neuroblasts adapt their migration modes based on their microenvironment, some of which are actively remodeled by the neuroblasts themselves. Given that the promotion of neuroblast migration enhances neurological recovery in mice after brain injury [8,19,20], understanding the underlying mechanism and biological significance of migration mode tuning of neuroblasts will help develop a new strategy for successful neuronal regeneration. However, this issue remains largely unexplored because of the technical challenges in consistently observing individual neuroblasts within a chain over an extended period and in manipulating neuroblast motility and environmental factors with precision.

Mathematical approaches can overcome these limitations because the determinants of long-term efficiency can be examined using a numerically constructed system with easily tunable individual rules. We integrated knowledge from in vitro and in vivo studies to build a mathematical model of the neuroblast-astrocyte system. Upon modeling, we placed importance on simplicity, which allows us to easily extract a key mechanism for the aspect in focus, in contrast to a complicated system in which numerous factors simultaneously affect each other. We then performed in silico simulations to understand how the collective behaviors of neuroblasts emerge and to visualize the long-term advantages of forming a collective. We found that neuroblasts in a collective migrate faster than solitary neuroblasts in conditions with densely populated astrocytes. Mathematical models are not merely an alternative to biological systems, but are superior in predicting new ideas, which might be biologically validated in the future. One major prediction from our simulations was that inconstancy (temporal change) in the movement of neuroblasts improves migration efficiency, which has received little attention in biological studies of cell migration.

## 2. Methods

We focused on several factors that have been shown to play critical roles in regulating neuroblast migration, such as adhesion between neuroblasts as well as their interactions with the surrounding cells. Based on these findings, we selected, for example, intercellular adhesion strength and astrocyte density as key parameters. We implemented these parameters into a mathematical model in which the neuroblast-astrocyte system was simplified on a two-dimensional field. In our assumption, neuroblasts are migratory agents (Sect 2.1); astrocytes are environmental agents (Sect 2.2); they make contact-based interactions (Sect 2.3). Briefly, the modeled neuroblasts moved forward via saltation cycles and adhered to each other in contact. Modeled astrocytes, which form a less accessible environment for neuroblasts, partially shrink in response to nearby neuroblast collectives, opening a migratory route. Using the mathematical model, we performed computer simulations with a set of parameter values (Sect 2.4) under several conditions (Sect 2.5) and compared the migration efficiency with the forward speed (Sect 2.6). This section ends with an ethics statement (Sect 2.7). The methods are detailed in S1 Appendix, the codes are available in S1 File, and the simulation results are presented in Sect 3.

### 2.1. Modeling a neuroblast

A typical migrating neuroblast has a soma with a long leading process (Fig 1). The tip of the process has a bushy shape with multiple filopodia [4,21]. Kinematically, neuroblasts follow a saltatory movement, which is a periodic sequence of resting and moving phases [17,21,22]. In the resting phase, the leading process extends forward while the soma is stationary (Fig 1a; 0–16, 40– min). This is followed by the moving phase in which the soma moves in the direction of the leading process, resulting in the contraction of the process (Fig 1a, 16–40 min).

In our model, we abstracted a neuroblast into a dumbbell shape—two differently sized circles connected by an elongated rectangle (Fig 2a). The large and small circles represent the "soma" and "tip," respectively. The rectangle represents the "process," which exhibits the characteristics of a spring linking the central coordinates of the soma and tip. Accordingly, the two-dimensional central coordinates of the $i$-th neuroblast's soma ($\boldsymbol{x}_{i,\mathrm{S}}(t)$) and tip ($\boldsymbol{x}_{i,\mathrm{T}}(t)$) ($i = 1, 2, \ldots N_{\mathrm{N}}$, number of neuroblasts) at time $t$ are updated as

$$\mu_{i,\mathrm{S}}(t)\frac{\mathrm{d}\boldsymbol{x}_{i,\mathrm{S}}(t)}{\mathrm{d}t} = k_{\mathrm{P}}\big(|\boldsymbol{x}_{i,\mathrm{S}}(t) - \boldsymbol{x}_{i,\mathrm{T}}(t)| - \bar{l}_i(t)\big)\boldsymbol{e}_{\mathrm{ST},i}(t) + \boldsymbol{F}_{i,\mathrm{S}}(t), \tag{1}$$

$$\mu_{i,\mathrm{T}}(t)\frac{\mathrm{d}\boldsymbol{x}_{i,\mathrm{T}}(t)}{\mathrm{d}t} = -k_{\mathrm{P}}\big(|\boldsymbol{x}_{i,\mathrm{S}}(t) - \boldsymbol{x}_{i,\mathrm{T}}(t)| - \bar{l}_i(t)\big)\boldsymbol{e}_{\mathrm{ST},i}(t) + \boldsymbol{F}_{i,\mathrm{T}}(t), \tag{2}$$

where $\mu_{i,\mathrm{S}}(t)$ and $\mu_{i,\mathrm{T}}(t)$ are friction coefficients of the soma and tip, respectively; $k_{\mathrm{P}} > 0$ is a spring constant of the process, with its target length defined by $\bar{l}_i(t)$ (target distance between the centers of the soma and tip circles, defined in the next paragraph); $\boldsymbol{e}_{\mathrm{ST},i}(t)$ is a unit vector from $\boldsymbol{x}_{i,\mathrm{S}}(t)$ to $\boldsymbol{x}_{i,\mathrm{T}}(t)$. Each $\boldsymbol{F}_{i,\mathrm{S}}(t)$ and $\boldsymbol{F}_{i,\mathrm{T}}(t)$ is the sum of the interactions with agents other than the $i$-th neuroblast (see Sect 2.3). Inertia is ignored because the frictional force dominates motion at the cell scale [23].

To express the cycle of saltatory movement, we periodically changed the spring's target length $\bar{l}_i(t)$ over time as follows:

$$\bar{l}_i(t) = L + A_{\mathrm{sal}}(t)\sin(\omega_{\mathrm{sal}}t + \phi_{\mathrm{sal},i}), \tag{3}$$

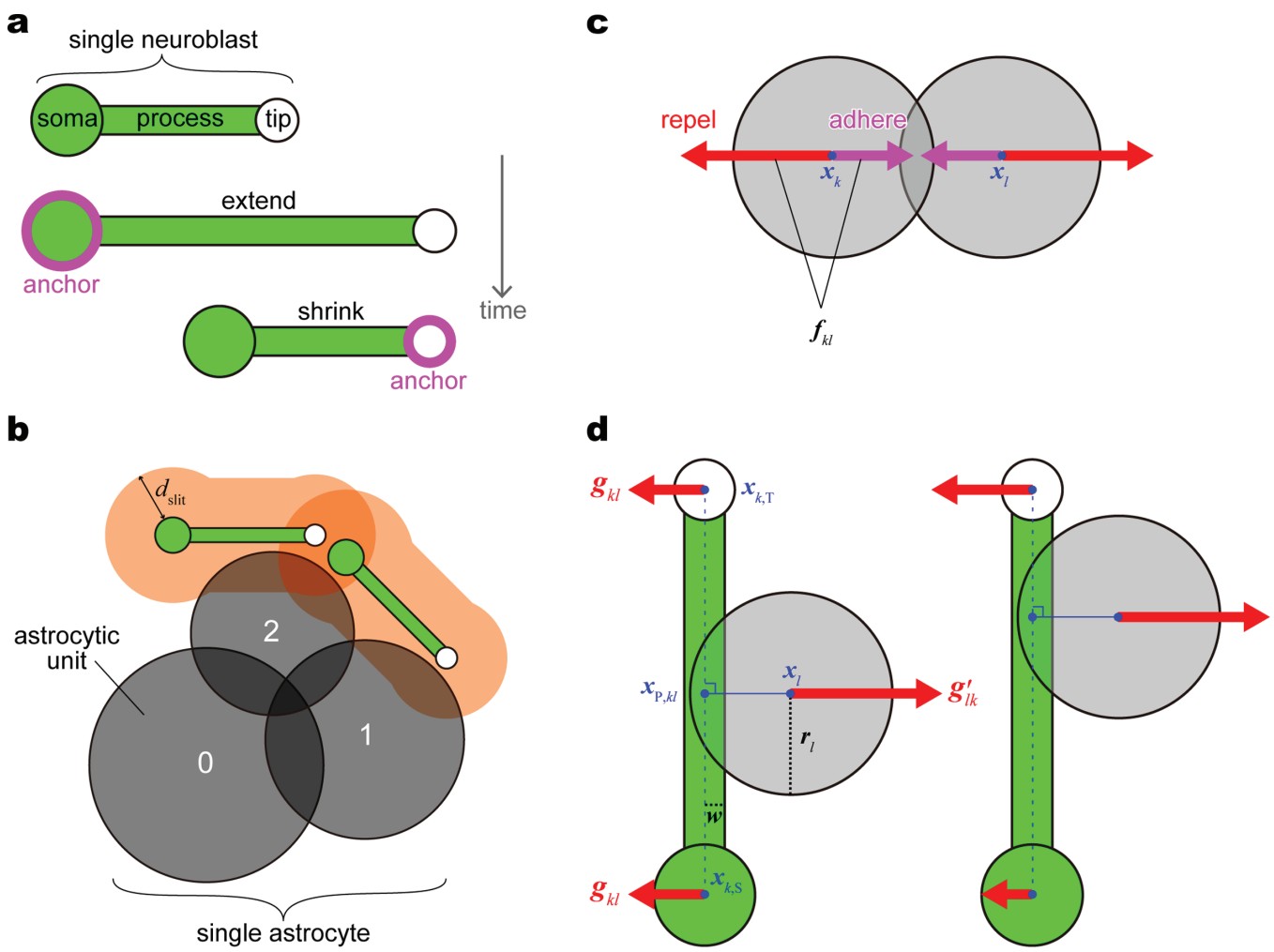

**Fig 2. Schematic of the mathematical model.** (a) Structure and saltation cycle of a single neuroblast. (b) Structure of a single astrocyte composed of three "astrocytic units," each of which shrinks depending on the number of neuroblasts within the distance $d_{slit}$ indicated by the orange shade ($B_{i,j}$, value shown at the center). (c) Repulsion and adhesion forces between two circles (soma, tip, or astrocytic unit) in contact. (d) Repulsion force between a circle (soma, tip, or astrocytic unit) and a neuroblast's process in contact. The arrow length represents the strength of the repulsive force. The left and right cases show the different force distributions to the soma and tip depending on the contact point $x_{P,kl}$; the definition is detailed in S1 Appendix (Sect IIa).

where $L$ and $\omega_{sal}$ are positive constants corresponding to the middle length and angular velocity of the saltatory movement of the spring, respectively, and $\phi_{sal,i}$ is the initial phase of the $i$-th neuroblast's saltation. The amplitude $A_{sal}(t)$ is either constant or temporally changed (see Sect 2.5iii).

The resting and moving phases are defined as the phases in which the target length $\bar{l}_i(t)$ increases and decreases, respectively. We assumed that an actual neuroblast in the resting phase anchors the soma to the extracellular matrix (ECM) to push the tip in the traveling direction; in the moving phase, the tip is anchored to the ECM to pull the soma (Fig 2a). We defined anchoring as a switch in the friction coefficients of the soma and tip ($\mu_{i,S}(t)$ in Eq 1 and $\mu_{i,T}(t)$ in Eq 2). The soma has a higher friction coefficient than the tip in the resting phase and vice versa in the moving phase:

$$(\mu_{i,\mathrm{S}}(t), \mu_{i,\mathrm{T}}(t)) = \begin{cases} (\mu_{\mathrm{low}}, \mu_{\mathrm{high}}) & \text{in the moving phase,} \\ (\mu_{\mathrm{high}}, \mu_{\mathrm{low}}) & \text{in the resting phase,} \end{cases} \mu_{\mathrm{high}} > \mu_{\mathrm{low}} > 0, \qquad (4)$$

which causes neuroblasts to migrate in the direction of the tip. We assume that a high friction coefficient corresponds to a strong adhesion to the ECM, as hypothesized earlier [21]. The actual neuroblast likely actively changes its moving direction depending on the environment [24,25]; however, for simplicity, the modeled neuroblast does not rotate or move backward unless it receives contact-based interactions with other agents (see Sect 2.3).

## 2.2. Modeling an astrocyte

We assumed that astrocytes, another type of cell, are the agents surrounding the neuroblasts. In vitro and in vivo studies have shown that neuroblasts secrete a diffusible protein, Slit1, that binds to the receptor protein Robo2 on astrocytes [8,18]. This signal causes astrocyte protrusions to retract at sites of contact with neuroblasts, thus apparently modifying the environment to clear the path toward the destination [8]. Based on biological evidence, our mathematical model includes the interaction with astrocytes, the outer environment for migratory cells, to investigate the function of neuroblast collectives in vivo.

We modeled a single astrocyte as three circularly connecting circles; hereafter, we refer to each circle as the "astrocytic unit" (Fig 2b). Although we did not assume the active migration of astrocytes, each astrocytic unit moved through contact-based interactions with other agents. Accordingly, the coordinates of the center of astrocyte $i$'s unit $j$, $\boldsymbol{x}_{i,j}(t)$, are updated as

$$\mu_{\mathrm{A}} \frac{\mathrm{d}\boldsymbol{x}_{i,j}(t)}{\mathrm{d}t} = \boldsymbol{F}_{i,j}(t), \qquad (5)$$

where $\mu_{\mathrm{A}} > 0$ is the friction coefficient of the astrocytic unit, and $\boldsymbol{F}_{i,j}(t)$ is the sum of the interactions—repulsion and adhesion—with objects other than the unit (see Sect 2.3).

We defined the partial retraction of astrocytic processes [8] as the shrinkage of an astrocytic unit near the neuroblasts. The minimum number of astrocytic units required to abstract the radially extended profile of an astrocyte capable of partial deformation was three. The radius of astrocyte $i$'s unit $j$ ($i = 1, 2, \dots N_{\mathrm{A}}$, number of astrocytes; $j = 1, 2, 3$) at time $t$ is updated as

$$\tau \frac{\mathrm{d}r_{i,j}(t)}{\mathrm{d}t} = \bar{r}_{i,j}(t) - r_{i,j}(t), \qquad (6)$$

where $\tau$ is a time constant related to the rate of the radius change to the target radius $\bar{r}_{i,j}$:

$$\bar{r}_{i,j}(t) = \max(r_{\mathrm{max}} - \rho_{\mathrm{A}} B_{i,j}(t), r_{\mathrm{min}}), \qquad (7)$$

where $B_{i,j}(t)$ is the number of neuroblasts within distance $d_{\mathrm{slit}}$ from the surface of the unit (Fig 2b). The target radius reaches the maximum $r_{\mathrm{max}}$ when there are no nearby neuroblasts, that is, $B_{i,j}(t) = 0$. The larger number of neuroblasts are close to the astrocytic unit, the smaller the target radius, unless it is below the minimum radius $r_{\mathrm{min}}$. The positive constant $\rho_{\mathrm{A}}$ determines how strongly the astrocytic unit $j$ shrinks when one more neuroblast appears within the vicinity.

## 2.3. Modeling inter-agent interactions

Any type of circles (soma, tip, or astrocytic unit) receives the interaction force summed from all circles in contact. We assumed mechanical repulsion, the excluded volume effect, between any two circles in direct contact (Fig 2c) or between circles in contact via a process (Fig 2d) in the direction away from each other. The strength of the repulsion is proportional to the overlapping length between two circles or between a circle and a process. In via-process repulsion, the force distribution to the soma and tip relies on the contact point on the process (Fig 2d).

In the direction opposite to direct-contact repulsion, we assumed adhesion between a certain combination of cell parts (Fig 2c). Based on the molecular studies, we considered intercellular adhesion via a chemical substance between neuroblasts [11] as well as between astrocytes [26]. To physically integrate the three astrocytic units into a single astrocyte, which should not split, we also assumed intracellular adhesion between the astrocytic units, which is substantially stronger than intercellular adhesion.

In the formula, the contact-based interaction forces $F_{i,S}(t)$, $F_{i,T}(t)$, and $F_{i,j}(t)$ in Eqs 1, 2, and 5, respectively, are commonly expressed by $F_k(t)$ acting on circle $k$ (soma, tip, or astrocytic unit):

$$F_k(t) = \sum_{l \in Z_{\text{CC},k}(t)} f_{kl}(t) + \sum_{l \in Z_{\text{PC},k}(t)} g_{kl}(t) + \sum_{l \in Z_{\text{CP},k}(t)} g'_{kl}(t). \tag{8}$$

The repulsion and adhesion forces between two circles in direct contact ($f_{kl}(t)$ in Fig 2c) are represented by the first term on the right-hand side; $Z_{\text{CC},k}(t)$ is a set of any circles (soma, tip, or astrocytic unit) in contact with circle $k$. The second and third terms on the right-hand side represent the repulsion forces via the process (Fig 2d). For example, when an astrocytic unit touches the process of a neuroblast, the soma and tip of the neuroblast shift according to the second term, and the astrocytic unit shifts according to the third term. Specifically, the second term expresses the force on the soma or tip $k$ against $Z_{\text{PC},k}(t)$, which is a set of other cell circles (soma, tip, or astrocytic unit) that contact the process extending from the soma or tip $k$ ($g_{kl}(t)$ in Fig 2d). The third term expresses the force on circle $k$ against $Z_{\text{CP},k}(t)$, which is a set of the somas and tips of other neuroblasts whose processes contact circle $k$ ($g'_{kl}(t)$ in Fig 2d). No contact is observed between the two processes for the parameter sets. S1 Appendix details each term (Sect IIa).

## 2.4. Parameter setting

Using the mathematical model proposed in Sects 2.1–2.3, computer simulations were performed in order to reproduce long-term in vivo neuroblast migration. We used biological knowledge [8,17,22] and our observation (Fig 1) to determine parameters for neuroblast morphology such as the average process length $L$ = 40 [μm], the average saltation amplitude $A_{\text{mid}}$ = 10 [μm], the radii of the soma and tip ($r_S$ = 5 [μm] and $r_T$ = 3 [μm], respectively), and half the width of the process ($w$ = 2 [μm]) (S1 Table). Based on the kinematics (Fig 1a), we set the wavelength of the saltation cycles to 20 min, that is, $\omega_{\text{sal}} = 2\pi/20$ [rad/min]. We determined the astrocyte size such that the width of the astrocyte territory was approximately 100 μm, tuning the maximum radius of astrocytic units ($r_{\text{max}}$) and the intracellular adhesion strength between astrocytic units ($H_a$).

We manually tuned the other parameters because of the technical difficulty in quantitative measurements (S1 Table). Upon manual tuning, we confirmed whether the visualized simulations agreed with the biological observations. To enable neuroblasts to alternately anchor the soma and tip for forward movement, smaller and larger friction coefficients ($\mu_{\text{low}}$ and $\mu_{\text{high}}$)

need to be sufficiently small and large, respectively. When neuroblasts collide with other neuroblasts, they hardly form a chain-like collective under a large value of the spring constant of the process ($k_P$) or small value of the intercellular adhesion strength between neuroblasts ($H_N$); in the opposite case, neuroblasts form a huge cluster incapable of displacement. We thus chose the intermediate values for these parameters to allow for chain formation with aspect ratios similar to those in vitro, i.e., the long axis three to four times longer than the short axis [22]. To keep astrocytes stationary even when pushed by neuroblasts, $k_P$ and the friction coefficient of astrocytic units ($\mu_A$) were set sufficiently small and large, respectively.

The intercellular adhesion strength between astrocytes ($H_A$) needs to be small enough for neuroblasts to pass between the astrocytes, but not zero given the biological basis [26]. The parameters related to the extent and rate of astrocyte shrinkage ($\tau$, $\rho_A$, $r_{min}$, and $d_{slit}$) were determined based on in vitro videos showing neuroblast-astrocyte interactions [8]. For example, astrocytes appear to retract their processes when neuroblasts are touching or close enough to touch, but not when neuroblasts are far away; we thus let the reachable distance of the Slit signal reasonably short ($d_{slit}$ = 15 [μm]). The coefficient for repulsion strength ($k_R$) was set large enough so that agents do not overlap or cross each other, but small enough so that astrocytic units within an astrocyte do not split. Because the manually tuned values are not biologically definitive, we investigated the robustness of the results using a range of parameter values to reinforce the reliance.

## 2.5. Simulation conditions

We distributed 24 neuroblasts and a certain number of astrocytes on a square field of 600×600 μm with a periodic boundary condition applied to the four sides. To determine the factors that influence the migration efficiency, we focused on i) the number of astrocytes, ii) whether astrocytes shrink in response to nearby neuroblasts, iii) the inconstancy in the properties of neuroblasts, and iv) the strength of adhesion between neuroblasts. Each of these aspects i)–iv) is explained in detail as follows.

i) Biological studies have shown that brain insults induce astrocyte proliferation, leading to the increased density of astrocytes—the activation state [4,27]. To test the influence of pathological changes in injured tissue, we compared simulations with different numbers of astrocytes in the field ($N_A$ = 0, 10, 20, 30, 40; Fig 3a and 3b). The large number of astrocytes makes it highly dense, representing the activation state.

ii) Living astrocytes retract the processes at sites in contact with neuroblasts via the Slit-Robo interaction [8,18]. We determined the significance of the Slit-Robo interaction by comparing the "reactive shrinkage" simulations, in which astrocytes shrink in response to nearby neuroblasts similarly to in vitro observations, and the "mismatched shrinkage" simulations, in which apparently random astrocytes shrink based on an unrealistic assumption. Because the total areas occupied by astrocytes become similar between the two cases, this comparison enables us to extract how significantly neuroblast migration in a certain space is affected by the realistic interaction between closely located neuroblasts and astrocytes. In the reactive shrinkage condition, we let each astrocytic unit shrink depending on the number of neuroblasts near the unit ($B_{i,j}(t)$ in Eq 7, Fig 3b). In contrast, in the mismatched shrinkage condition, when a neuroblast approaches a certain astrocytic unit, a randomly chosen unit instead of the nearby unit shrinks (Fig 3c).

iii) The migration of V-SVZ-derived neuroblasts is controlled by multiple factors that change the amount in response to, for example, food intake and the circadian rhythm

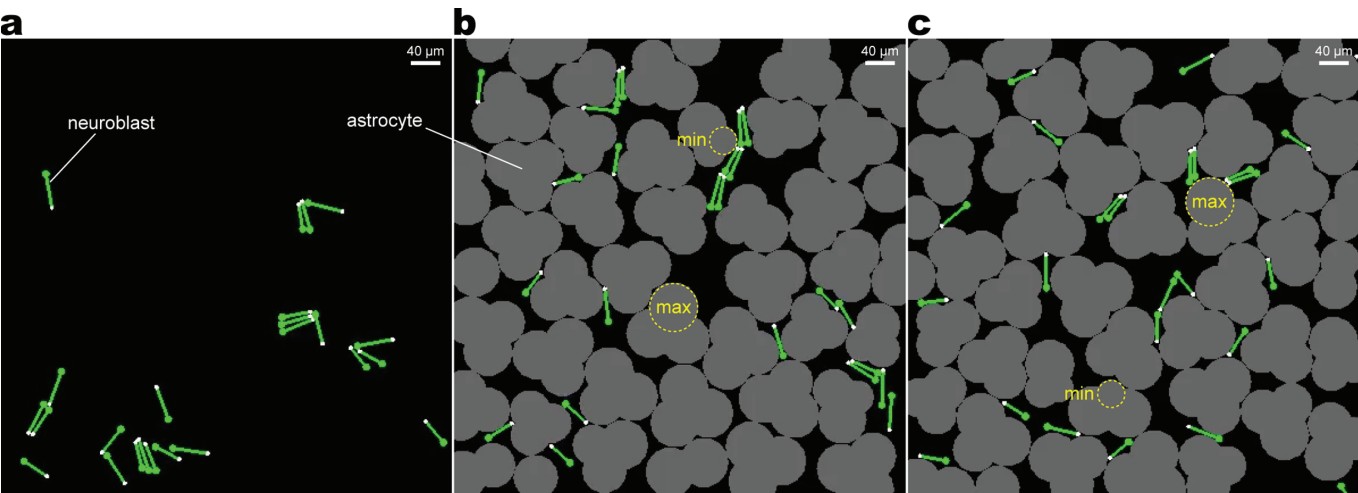

**Fig 3. Simulation fields.** (a) Without astrocytes ($N_A$ = 0). (b) With "reactive shrinkage" astrocytes ($N_A$ = 40). An astrocytic unit with many nearby neuroblasts targets the minimum radius ($r_{min}$, dotted circle with "min"), while that without nearby neuroblasts targets the maximum ($r_{max}$, dotted circle with "max"), imitating the Slit-Robo interaction as defined in Eq 7. (c) With "mismatched shrinkage" astrocytes ($N_A$ = 40). Since astrocytic units shrink pseudorandomly, a unit without nearby neuroblasts may target the minimum radius (dotted circle with "min"), while that with many nearby neuroblasts may target the maximum (dotted circle with "max")—an unrealistic rule to compare with the reactive shrinkage. Each snapshot is captured from a late stage of the simulation ($t$ = 6300 [min]).

[28–30]. Additionally, our in vitro tracking of a neuroblast showed a variation in the speed of forward movement through a time series (Fig 4a; methods detailed in Sect I of S1 Appendix). Therefore, we further hypothesized that migration efficiency is improved by inconstancy (temporal change) in the movement of neuroblasts. To test this, we compared the constant and inconstant cases in $A_{sal}(t)$, the amplitude of the target length in the saltatory movement (Eq 3). In the inconstant case, we let $A_{sal}(t)$ change according to a common long-term activity, which is either periodic (Fig 4b, green curve) or random (Fig 4b, blue curve). The periodic conditions were expressed as

$$A_{sal}(t) = A_{mid} + A_{act}\sin(\omega_{act}t + \phi_{act}),\qquad(9)$$

where the amplitude $A_{act}$ = 5 [μm], angular velocity $\omega_{act} = \omega_{sal}/36$, initial phase $\phi_{act}$ = $3\pi/2$, and middle saltation amplitude $A_{mid}$ = 10 [μm]. As an alternative to Eq 9, $A_{sal}(t)$ in the random conditions follows the normal distribution symbolized by $\mathcal{N}$:

$$A_{sal}(t) \sim \mathcal{N}(A_{mid}, \sigma_A),\qquad(10)$$

with the mean $A_{mid}$ = 10 [μm] and standard deviation $\sigma_A$ = 5 [μm]. The random values were truncated within [0,20] and updated every $u$ saltation cycles; we let $u$ = 3. The average of $A_{sal}(t)$ is theoretically equivalent to that in the constant case (Fig 4b, black curve), in which a common amplitude is constantly set for all neuroblasts, i.e., $A_{sal}(t) = A_{mid}$ = 10 [μm]. For the supplementary simulations, we applied other values of the inconstancy-related parameters in periodic ($A_{act}$ and $\omega_{act}$) and random conditions ($\sigma_A$ and $u$). In addition to inconstancy in $A_{sal}(t)$, we tested the inconstancy in adhesion strength between neuroblasts ($H_N(t)$) to confirm if the migration efficiency is improved by a different type of inconstancy.

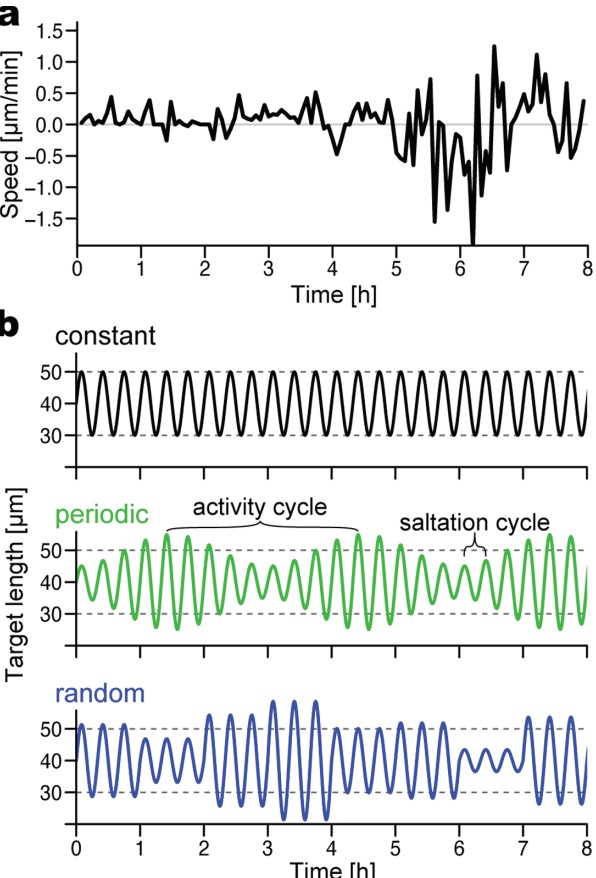

**Fig 4. 'Activity' of neuroblasts.** (a) Time series of the speed of a neuroblast in vitro (methods detailed in Sect I of S1 Appendix). (b) Target length of a neuroblast process ($\bar{l}_i(t)$ in Eq 3) in silico. The wavelength of saltation cycles is set to 20 min, i.e., $\omega_{sal} = 2\pi/20$ [rad/min] in Eq 3. The saltation amplitude ($A_{sal}(t)$) depends on the simulation conditions: constant (black; $A_{sal}(t) = 10$ [μm]), periodic (green; the case of nine saltation cycles per activity cycle, i.e., $\omega_{act} = \omega_{sal}/9$ and the activity amplitude $A_{act} = 5$ in Eq 9), and random (blue; the case of the saltation amplitude under the truncated normal distribution with the mean $A_{mid} = 10$ and the standard deviation $\sigma_A = 5$ in $0 \leq A_{sal}(t) \leq 20$, updated every three saltation cycles, i.e., $u = 3$). Average target lengths through a time series in all conditions theoretically equal the middle length $L = 40$ [μm]. Dotted lines indicate the minimum and maximum lengths under the constant case (30 and 50 μm, respectively).

iv) Neuroblasts adhere to each other to form a collective [11]. Supposing that intercellular adhesion is a key factor in chain formation, we tested the influence of the parameter $H_N$, representing the adhesion strength between neuroblasts. Although we set $H_N = 2.22$ [nN] in the above-mentioned simulations (except for the inconstant adhesion conditions), we applied other constant values from 0 through 3.33 nN.

In a condition with the same parameter set, we performed 10 trials with random initial positions for the agents. Each trial had a duration equivalent to 9,000 min (150 h, 6.25 days) in real time. S1 Appendix details the simulation conditions (Sects IIb–IId).

## 2.6. Evaluation of migration efficiency

To compare the migration efficiency under different simulation conditions, we measured the "forward speed" of each neuroblast at time intervals of $t_{eval}$ = 60 [min]. The speed of neuroblast $i$ in a simulation trial was represented by

$$v_i(t) = \frac{e_{ST,i}(t) \cdot (x_{i,S}(t) - x_{i,S}(t - t_{eval}))}{t_{eval}}, \qquad (11)$$

where $e_{ST,i}(t)$ is the unit vector from the soma to the tip, and $x_{i,S}(t)$ is the coordinate of the soma at time $t$ (Fig 5a). When a neuroblast moves straight along the soma-to-tip axis, $v_i(t)$ is equivalent to the velocity of the soma per minute and is negative during backward movement. The velocity was weighted less when it moved diagonally along the soma-to-tip axis.

We hypothesized that the forward speed is influenced by the degree of collective formation. Such an explanatory variable termed "collectivity," $n_i(t)$, was defined as the number of neuroblasts nearby neuroblast $i$ within a distance of $d_{near}$ = 20 [μm] from the surface (Fig 5b). In each trial of a simulation condition, the pair of forward speed $v_i(t)$ and collectivity $n_i(t)$ (Fig 5c) was calculated for as many as 24 neuroblasts multiplied by 136 evaluation time points (60 min intervals from 900 to 9,000 min).

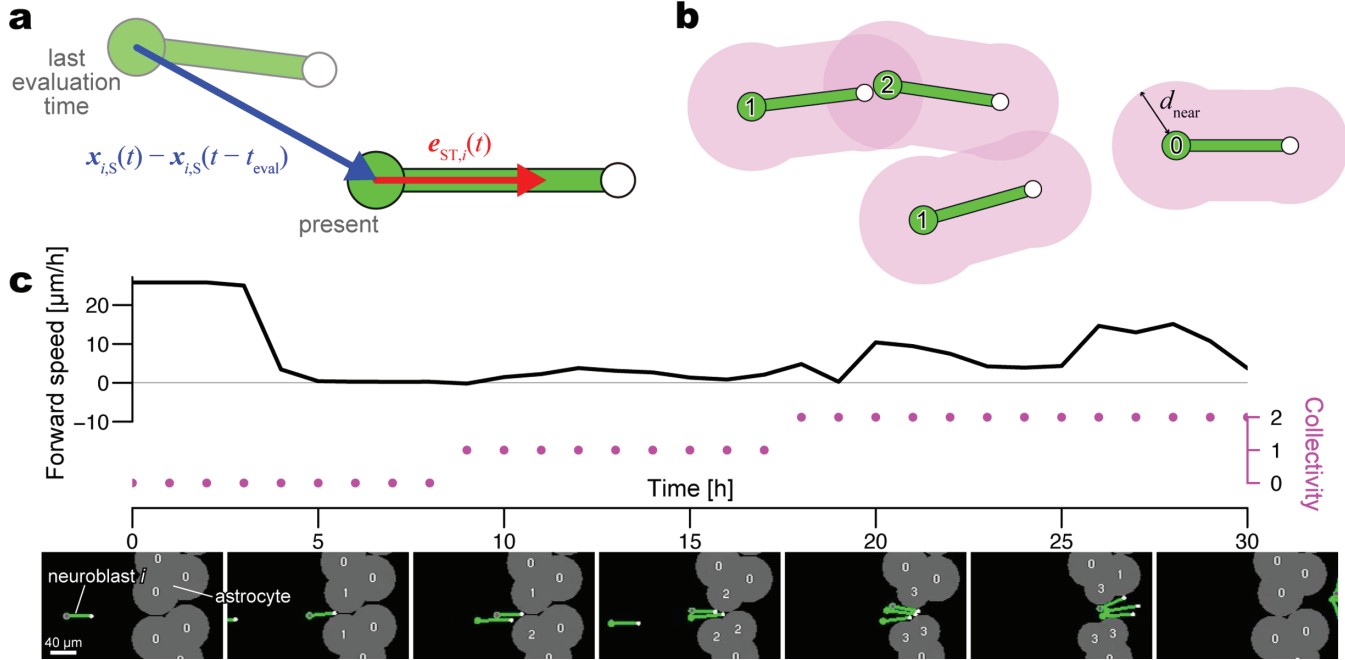

**Fig 5. Evaluation of simulated migration.** (a) Measurements for calculating the 'forward speed' ($v_i(t)$ in Eq 11) for a neuroblast. The large green and small white circles represent the soma and tip, respectively. The migration distance per hour (blue arrow length, $t_{eval}$ = 60 [min]) is weighted by the extent to which the soma translocates in the tip direction. (b) Number of neuroblasts within a distance ($d_{near}$, magenta shade) for a neuroblast, namely, 'collectivity' ($n_i(t)$). The number at each soma indicates the collectivity value. (c) Forward speed (solid line, left axis) and collectivity (dotted line, right axis) of a neuroblast through a time series in a preliminary simulation. Each dot of the collectivity represents an evaluation time point. Snapshots at seven time points show that three modeled neuroblasts pass between two astrocytes; the preceding neuroblast with a point labeled at the soma is neuroblast $i$ for the plot. The value at the center of each astrocytic unit indicates the number of nearby neuroblasts ($B_{i,j}(t)$ in Eq 7), determining the target radius ($\bar{r}_{i,j}(t)$).

To elucidate how the number of nearby neuroblasts explains the migration efficiency, the relationship from the collectivity to forward speed was fitted with a linear regression:

$$v_i(t) \sim \alpha n_i(t) + \beta. \tag{12}$$

A positive regression slope $\alpha$ indicates that a neuroblast in a collective tends to move forward faster than a solitary neuroblast, whereas a negative value indicates the opposite tendency. The regression intercept $\beta$ indicates the forward speed when a neuroblast is solitary, that is, $n_i(t) = 0$. For each simulation condition, we used all the data from 10 trials to obtain a single regression line.

## 2.7. Ethics statement

All experiments involving live animals were approved by the Doshisha University Institutional Animal Experiment Committee (approval number: A24065) and were performed in accordance with the guidelines of the committees.

## 3. Results

In silico, neuroblasts migrated via saltatory movements and were rotated by physical contact with other agents. Although neuroblasts were initially scattered and faced different directions, occasionally colliding neuroblasts formed a collective and often became parallel to each other (Fig 6a and 6b), regardless of the simulation conditions. Neuroblasts in a collective sometimes faced multiple directions (Fig 6c and 6d) and separated into solitary neuroblasts (Fig 6d, 12h). Several simulation conditions yielded large positive values of the regression slope $\alpha$, that is, a neuroblast in a collective tended to migrate faster than a solitary one. In summary, the collective became markedly advantageous when astrocytes were densely occupied (Sect 3.1), when astrocytes shrank in response to nearby neuroblasts (Sect 3.2), when neuroblasts were inconstant in saltation amplitude (Sect 3.3), and when neuroblasts moderately adhered to each other (Sect 3.4).

## 3.1. High-density astrocytes created a collective advantage

When reactive-shrinking astrocytes densely occupied the simulation field, a solitary neuroblast shrank the surrounding astrocytes slightly, which was usually insufficient to open a route to pass through (Fig 6a, 0–48 h; S2 Video). Solitary cells were thus frequently stranded against astrocytes, resulting in small positive or negative forward speed values at a collectivity of 0. Once a neuroblast aggregated with others, the surrounding astrocytes shrank in proportion to the number of nearby neuroblasts (Fig 6a, 51–63 h, asterisks; S2 Video). The collective frequently opened a migration route to proceed, reflecting large positive values of forward speed with large collectivity. Consequently, neuroblasts in a collective were faster than a solitary neuroblast in a pool of astrocytes, as represented by the positive values of the regression slopes (Fig 7a, $N_A = 30, 40$; Fig 7b, triangles/crosses; S3 Video).

The collectives were robustly advantageous in a range of parameter variations, such as the spring constant of a neuroblast process $k_P$, the time constant for astrocyte shrinkage $\tau$, the extent of astrocyte shrinkage determined by the number of nearby neuroblasts $\rho_A$, the distance of the Slit signal $d_{\text{slit}}$, and the adhesion strength between astrocytes $H_A$ (S1 Fig, positive slope values in the 'constant' columns). Moreover, the advantage remained consistent when each neuroblast was smoothened into a rounded rectangle ($r_S = r_T = w$) and when each

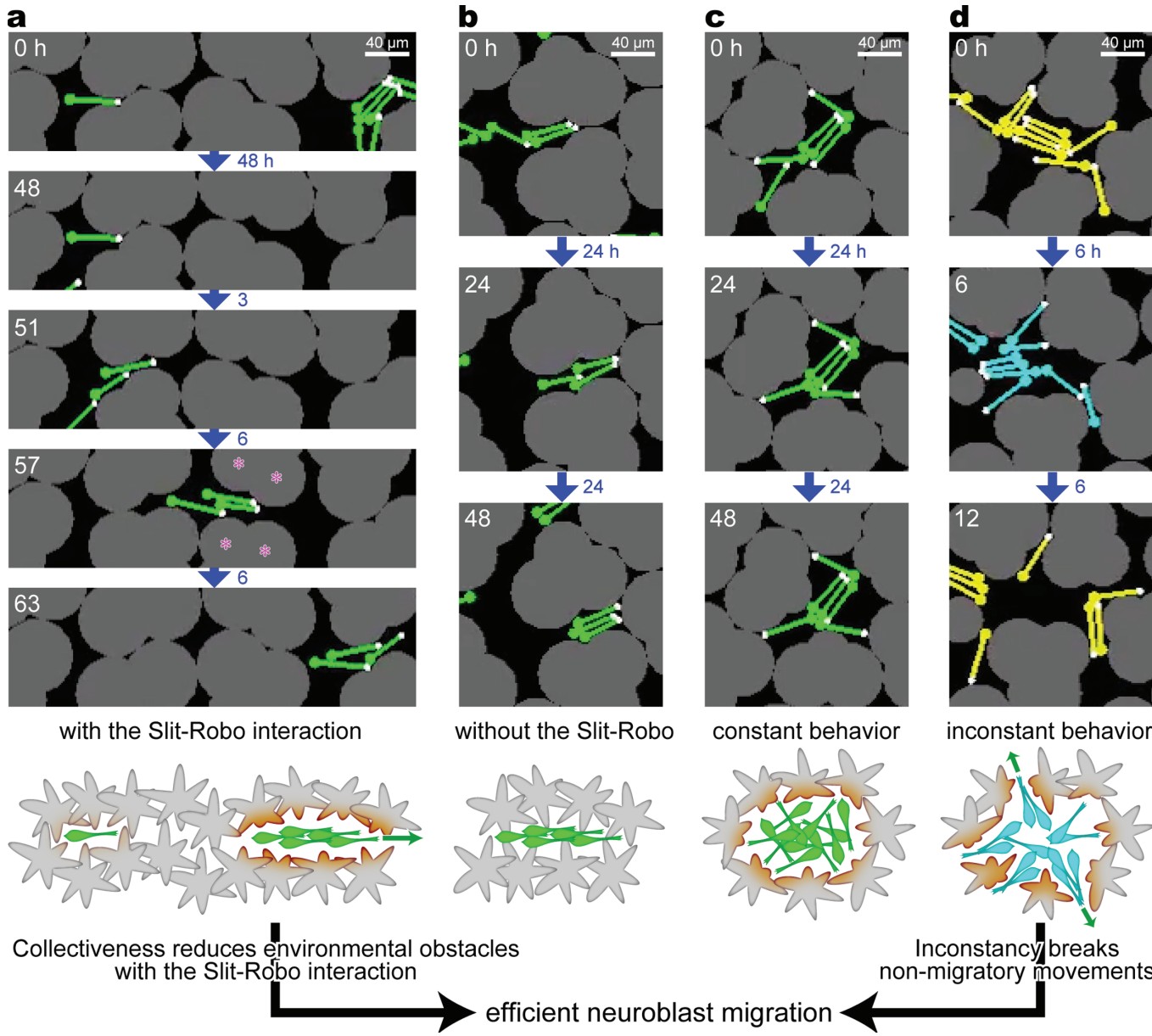

**Fig 6. Snapshots of simulations related to the shrinkage of astrocytes and the inconstancy of neuroblasts.** Total number of astrocytes $N_A$ = 40, with reactive shrinkage in (a, c, and d) but mismatched shrinkage in (b). The saltation amplitude of neuroblasts is constant in (a–c) but periodically changed in (d), all with the adhesion strength $H_N$ = 2.22 [nN]. (a) Scene when a solitary neuroblast is stuck for a long time and then joins with two others to efficiently open a migration route. Asterisks at 57 h show an example of strongly shrinking astrocytic units that contribute to the opening of a migration route. (b) Scene when a chain is stuck for a long time since a collective does not necessarily open a migration route under the mismatched shrinkage condition. (c) Scene when a cluster of several neuroblasts is stuck in a repetitive movement for a long time under the reactive shrinkage condition. (d) Scene when a cluster of several neuroblasts breaks a repetitive movement in a short time, leading to a high forward speed at a high collectivity. The wavelength of the activity cycle is set to 12 h, i.e., $\omega_{sal}/\omega_{act}$ = 36 (yellow, minimum; cyan, maximum). The time inside each snapshot indicates the elapsed time from the uppermost snapshot. The time next to the arrow indicates the interval between two continuous snapshots. The lowermost illustrations depict the advantage and disadvantage of collectivity; the latter is reduced by the inconstancy of neuroblast movements. The deep orange color indicates parts of astrocytes that receive the signal of nearby neuroblasts and thus retract.

astrocyte was simplified into an exact circle with a single astrocytic unit (S2 Fig), thus indicating that the frictional effect of the agents' concave and convex profiles was negligible on the results.

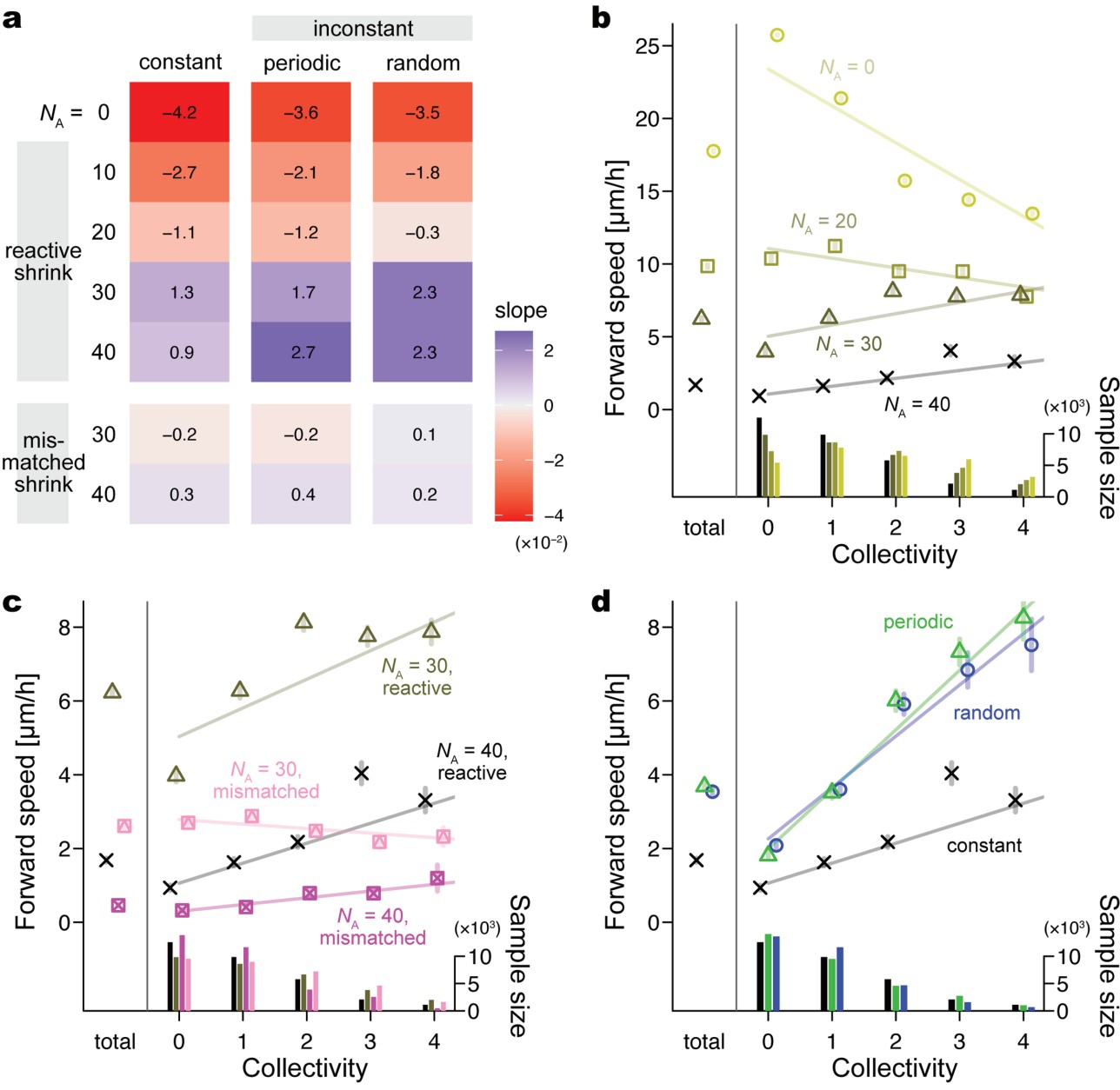

**Fig 7. Simulation results related to the number and shrinkage of astrocytes and the inconstancy of neuroblasts.** (a) Heat map of the regression slope $\alpha$ of the forward speed $v_i(t)$ for the collectivity $n_i(t)$ (the number of neuroblasts near each neuroblast). A large positive value on a blue background indicates that a neuroblast in a collective tends to be faster than a solitary one; a large negative value on a red background represents the opposite tendency. Values on the left side indicate $N_A$, the number of astrocytes. Upper rows labeled "reactive shrink" show conditions in which astrocytic units shrink depending on the number of nearby neuroblasts. Lower rows labeled "mismatched shrink" show the conditions where apparently random astrocytic units shrink. The left column indicates the conditions where neuroblasts have a constant saltation amplitude, that is, $A_{act} = 0$ (Eq 3). The middle column indicates periodic-inconstant conditions with $A_{act} = 5$ and $\omega_{sal}/\omega_{act} = 36$ (Eqs 3, 9). The right column indicates random-inconstant conditions with $\sigma_A = 5$ and $u = 3$. (b–d) Averaged plots of the forward speed for each number of collectivity. Symbol, mean; error bar, 95% confidence interval; line, regression line. An upward line to the right (i.e., positive slope) indicates that a collective tends to be faster than a solitary. Each leftmost symbol labeled "total" represents the mean overall collectivity values. Histograms at the bottom indicate the sample sizes from the above-plotted conditions. (b) Plots over different numbers of astrocytes, all with constant neuroblasts and reactive shrinkage. Cross, $N_A = 40$; triangle, $N_A = 30$; square, $N_A = 20$; circle, $N_A = 0$. (c) Plots to compare reactive versus mismatched shrinkage, all with constant neuroblasts. Cross, $N_A = 40$, reactive shrinkage; triangle, $N_A = 30$, reactive shrinkage; square-cross, $N_A = 40$, random shrinkage; square-triangle, $N_A = 30$, random shrinkage. (c) Plots to compare constant versus inconstant neuroblast movement, all with $N_A = 40$ and reactive shrinkage. Cross, constant; triangle, periodic-inconstant, $A_{act} = 5$ and $\omega_{sal}/\omega_{act} = 36$; circle, random-inconstant, $\sigma_A = 5$ and $u = 3$. Adhesion strength between neuroblasts is constant, i.e., $H_N = 2.22$ [nN], in all the conditions.

Simulations without astrocytes or with sparse astrocytes showed the opposite tendency. A solitary migrated faster than a collective one, as reflected by the negative values of the regression slopes (Fig 7a, $N_A$ = 0, 10, 20; Fig 7b, circles/squares; S3 Video). This indicates that the collective formation disturbed migration when neuroblasts received little or no physical inhibition from the environment.

## 3.2. Astrocyte shrinkage near neuroblasts created a collective advantage

When mismatched astrocytes shrank independently of the neuroblast positions, a neuroblast collective did not necessarily shrink astrocytes in the front and frequently became stuck as a solitary neuroblast (Fig 6b, S2 Video, and S3 Video). As a result, the advantage of forming a collective was poor (Fig 7a, bottom; Fig 7c, square-crosses/square-triangles). Given the collective-advantageous tendency of the normal simulations with reactive shrinkage (Fig 7c, crosses/triangles), we suppose that the advantage of collectivity was improved by the practical neuroblast-astrocyte interaction, by which the degree of shrinkage of astrocytes is proportional to the number of nearby neuroblasts.

## 3.3. Inconstant neuroblast behavior created a collective advantage

A collective composed of neuroblasts with a constant saltation amplitude sometimes showed repetitive movement without translocation for a certain period (Fig 6c and S2 Video). In contrast, the inconstant saltation amplitude frequently triggered a different movement, characterizing the easily deformable property (Fig 6d and S2 Video). In additional simulations in which all of the neuroblasts were initially clustered at the center of the field, the neuroblast cluster reduced collectivity when the saltation amplitude was large (S3 Fig). Consequently, the collective-advantageous tendency became evident when neuroblasts had inconstant saltation amplitude while reactive-shrinking astrocytes were dense ($N_A$ = 30, 40) (Fig 7a).

The advantage of inconstancy was almost consistent, regardless of the width and frequency of the saltation amplitude changes. Under the conditions with 40 astrocytes, inconstant cases with various widths ($A_{act}$ or $\sigma_A$) and time periods ($\omega_{sal}/\omega_{act}$ or $u$) in amplitude change mostly resulted in larger positive regression slopes than the constant case with a slope $\alpha$ of 0.9 (S4 Fig). In particular, wide-amplitude changes yielded large slopes under both periodic and random conditions (e.g., $A_{act}$ = 10 and $\sigma_A$ = 5, respectively). Although the tendency along the time periods ($\omega_{sal}/\omega_{act}$ or $u$) was not clear, infrequent amplitude changes in the random conditions, similar to the constant conditions, tended to show small slopes (e.g., $u$ = 36, 72 in S4 Fig). A range of parameter values mostly resulted in the advantage of collectives with the inconstant saltation amplitude compared to those with the constant amplitude (S1 Fig and S2 Fig), indicating the robustness of the results.

Moreover, the migration efficiency was improved by different types of inconstancy. Collectives with inconstant adhesion strength between neuroblasts but constant amplitude demonstrated a similar advantage to those with inconstant amplitude and constant adhesion strength (S5 Fig).

## 3.4. Moderate adhesion between neuroblasts created a collective advantage

Without adhesion ($H_N$ = 0 [nN]), the collective followed a chain shape in narrow areas between astrocytes (Fig 8a, 0–6 h) but easily dispersed in unconfined areas (Fig 8a, 6–10 h;

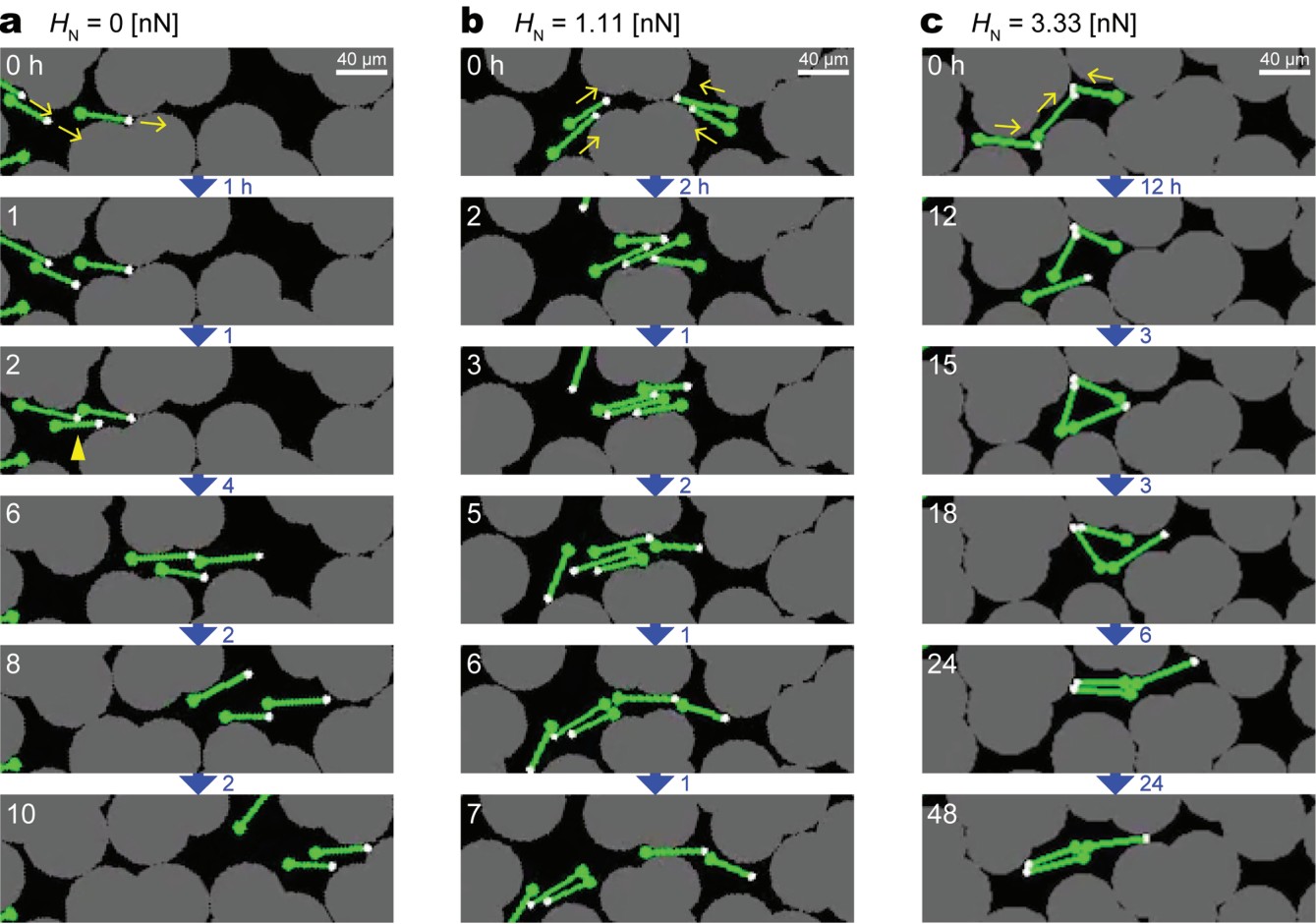

**Fig 8. Snapshots of simulations with different constant values of $H_N$, the adhesion strength between neuroblasts.** (a) $H_N = 0$ [nN], (b) $H_N = 1.11$ [nN], and (c) $H_N = 3.33$ [nN]. In all the conditions, the total number of astrocytes is $N_A = 40$ with reactive shrinkage, and the saltation amplitude of neuroblasts is constant. The time inside each snapshot indicates the elapsed time from the uppermost snapshot. The time between two continuous snapshots indicates the interval. The yellow arrows in each first snapshot indicate the forward directions of neuroblasts. The arrowhead in (a) at 2 h indicates that the neuroblasts form a chain-like collective with the support of the lower astrocyte.

S4 Video and S5 Video). The tendency to scatter caused the solitary neuroblasts to have the highest frequency among the conditions with varied adhesion strengths (Fig 9b, histogram at collectivity 0). The above-mentioned situation also indicates that astrocytes helped neuroblasts maintain a collective form even without adhesion (Fig 8a, 1–2 h, arrowhead).

In contrast, the stronger the adhesion, the more frequently the neuroblasts formed a large cluster (Fig 9b, histograms; S5 Video). Although a large cluster had the potential to significantly shrink the surrounding astrocytes, excessive adhesion often made the cluster stationary by interfering with the forward movement of each neuroblast, especially when the collective included oppositely facing neuroblasts (Fig 8c, $H_N = 3.33$ [nN]). Accordingly, a collective reduced the advantage in forward speed under excessive adhesion conditions (Fig 9a, bottom; Fig 9b, square-cross).

A moderate adhesion strength helped maintain a chain-like shape, even in unconfined areas. In this case, the following neuroblasts moved in a direction that touched the leading

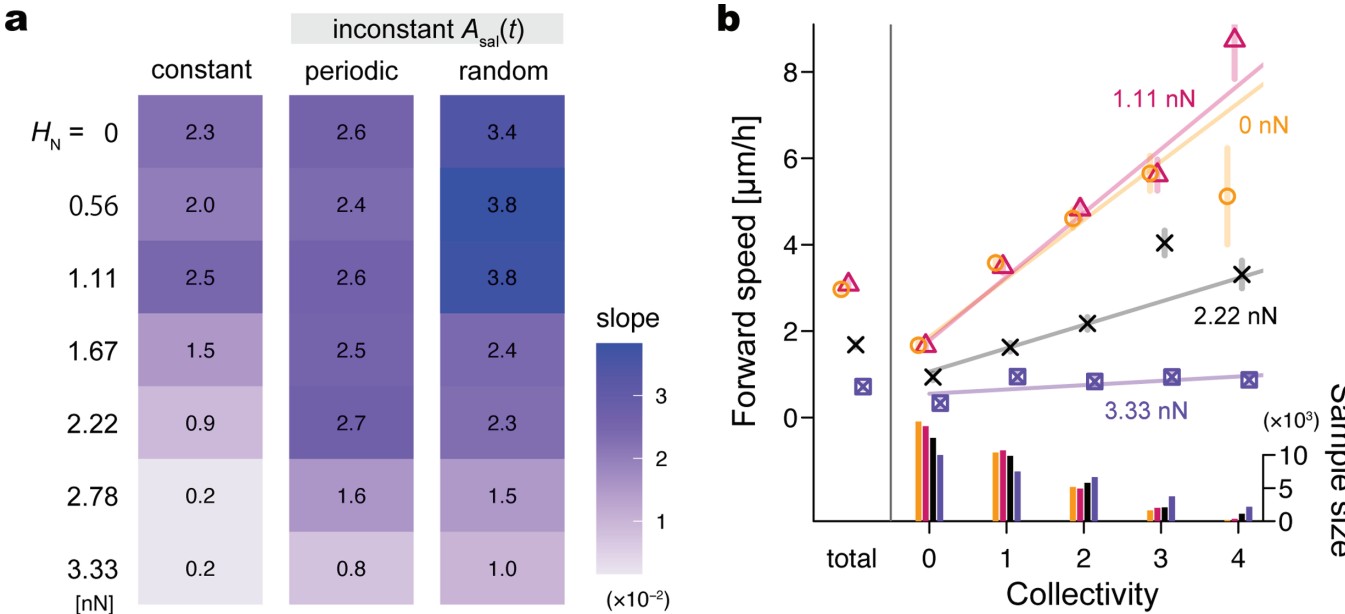

**Fig 9. Simulation results for various adhesion strengths between neuroblasts.** (a) Heat map of the regression slope $\alpha$ of the forward speed $v_i(t)$ for the collectivity $n_i(t)$ (the number of neuroblasts near each neuroblast). A large positive value on a blue background indicates that a neuroblast in a collective tends to be faster than a solitary one. Values on the left side indicate $H_N$, a constant adhesion strength between neuroblasts. The three columns indicate constant, periodic-inconstant, and random-inconstant conditions in saltation amplitude from the left. The parameters are as set in Fig 7. (b) Averaged plots of the forward speed for each number of collectivity over different values of $H_N$ with constant neuroblasts. Symbol, mean; error bar, 95% confidence interval; line, regression line. An upward line to the right (i.e., positive slope) indicates that a collective tends to be faster than a solitary. Each leftmost symbol labeled "total" represents the mean overall collectivity values. Histograms at the bottom indicate the sample sizes from the above-plotted conditions. Circle, $H_N$ = 0 [nN]; triangle, $H_N$ = 1.11 [nN]; cross, $H_N$ = 2.22 [nN]; square-cross, $H_N$ = 3.33 [nN]. The number of astrocytes showing reactive shrinkage is $N_A$ = 40.

neuroblasts, maintaining gentle adhesive binding over time (Fig 8b, $H_N$ = 1.11 [nN]). Meanwhile, oppositely facing neuroblasts tended to smoothly pass each other because the adhesion force was not sufficiently large to interfere with the forward-moving force. Compared with the no-adhesion cases, the slope $\alpha$ was slightly larger when $H_N$ took moderately large values; we found the largest slopes at 1.11 nN in the constant, 2.22 nN in the periodic, and 0.56 nN in the random conditions in saltation amplitude (Fig 9a). In addition, as in Sect 3.3, collectives with inconstant amplitudes are consistently advantageous, even when the adhesion strength took other constant values.

## 4. Discussion

In the neuroblast migration model, we propose a mechanism in which individual cells form a chain (Sect 4.1), and a collective of cells migrates faster than a single cell (Sect 4.2). We then discuss how a collective with inconstant cells is more advantageous than one with constant cells (Sect 4.3). Finally, we conclude the paper with future perspectives (Sect 4.4).

### 4.1. Mechanism of chain formation

In the brain, migrating neuroblasts are guided by various diffusible and non-diffusible factors supplied by surrounding cells and circulating fluids (e.g., blood and cerebrospinal fluid). In contrast, because our model did not impose any directional control on the neuroblast movement (see Sect 4.4), the alignment between colliding neuroblasts in the simulations emerged

passively through physical interactions between their elongated bodies. Whether the neuroblasts afterward face the same direction (aligned; cf. Fig 8b, 5–7 h, left group) or opposite directions (anti-aligned; cf. Fig 8b, 0–3 h) depends primarily on their contact angle, such as the collision of swimming filamentous bacteria [31]. Astrocytes bordering a narrow path also contribute to the alignment of neuroblasts facing divergent directions (Fig 8a, 1–2 h, arrowhead). This reinforces the suggestion from a previous biological study that astrocytes may provide a physical boundary lining the migratory route of neuroblasts [11]. Spatial confinement that leads to a high degree of alignment has been modeled in other systems, such as high-density filamentous bacteria in a confined colony [32] and a bundle of worms in a narrow channel [33].

The adhesion strength between neuroblasts is a factor that affects the stability and flexibility of the collective behavior after physical contact (Fig 8). Previous studies have found that multiple cell adhesion molecules, including N-cadherin, PSA-NCAM, and β1 integrin, are involved in the adhesion between neuroblasts [14,15], and the adhesion strength $H_N$ is thus converted to the expression level of such molecules. Interestingly, moderate adhesion associated cells moving in similar directions and dissociated cells moving oppositely (Fig 8b). Non-adhesive cells tended to scatter owing to poor associability (Fig 8a), while over-adhesive cells tended to stall owing to poor dissociability (Fig 8c). Our results agree with those of a recent study that demonstrated the advantage of appropriate neuroblast adhesion levels in chain migration using a mathematical model [19]. We note the importance of the balance between the intercellular adhesion force of neuroblasts and the forward-moving force of each neuroblast. The forward-moving force is determined by the spring constant of the process, and microtubule organization [21,34], and the friction coefficients for anchoring part of the cell, which are related to ECM-adhesive molecules such as laminin [11]. Taken together, our results suggest that the formation of chain-like collectives is mediated by occasional contact between migrating neuroblasts, narrow confinement by the surrounding astrocytes, and moderate adhesion between neuroblasts.

## 4.2. Advantage of collectivity

Neuroblast-astrocyte interaction, the retraction of astrocytic processes in response to the neuroblast-derived diffusible protein Slit1 [8], was incorporated into our model as a possible interaction between neuroblasts and the environment (Fig 2b). The maximum migration speed in the collective neuroblasts was relatively faster than that in solitary neuroblasts in a field with densely populated astrocytes, but not with sparse or no astrocytes (Fig 7b). In contrast, the immobility and slow migration of neuroblasts in mismatched shrinkage simulations (Fig 6b) resemble Slit1-knockout neuroblasts in vitro and in vivo [8]. Therefore, in addition to the involvement of Slit1 in interactions with astrocytes for the efficient migration of neuroblasts as demonstrated in the previous studies [8,18], in silico experiments newly indicate the significance of collective migration in this context (Fig 7c). Because astrocyte activation is generally observed under various pathological conditions [4,27], the collective migration of neuroblasts may be beneficial for the efficient distribution of new neurons in the damaged region for neurological recovery.

Migrating neuroblasts actively remodel not only astrocytes but also other microenvironments. V-SVZ-derived neuroblasts secrete matrix metalloproteinases that degrade ECM components [35]. Notably, neuroblasts form chain-like collectives in high concentrations of ECM-rich hydrogels but not at low concentrations [36], suggesting the advantage of collectivity in a dense ECM. We thus interpret the shrinkage of the 'astrocytic units' in our simulations as an analogy to the degradation of the ECM proteins. These results suggest that the primary

requirement for a collective to be advantageous is the presence of environmental objects that physically obstruct neuroblast migration (Fig 7b).

## 4.3. Advantage of inconstancy

Our in silico study comparing constant and inconstant saltatory amplitude conditions indicates that inconstant behavior in collective migration has a 'mechanical advantage.' Large neuroblast clusters occasionally became stationary in the fields with accumulated astrocytes. While the average saltatory amplitude of neuroblasts throughout the examination period was theoretically the same as that under constant conditions, temporary large-amplitude movements contributed to deadlock breaking (S3 Fig), which increased the efficiency of the forward movement of the large neuroblast clusters (Fig 6d). Inconstancy in adhesion strength between neuroblasts also showed a collective-advantageous tendency (S5 Fig), probably because the resulting movements of neuroblasts became inconstant, similarly to the case of inconstant saltation amplitude. For example, a neuroblast process at a weak-adhesion timing easily followed the intrinsic target length without interference by adhesion, causing an increase in the resulting amplitude. To the best of our knowledge, the advantage of inconstant behavior provides a novel insight into collective migration.

Both periodic and random changes in neuroblast properties increased the forward movement of neuroblasts in large collectives, but not of solitary migrating neuroblasts (Figs 7a, 7d, 9a, S4 Fig, and S5 Fig). Physiologically, the V-SVZ-derived neuroblast migration in the postnatal brain is controlled by multiple hormones, neurotransmitters, and neurotrophic factors [28–30]. In general, these factors are transiently released in response to stimulations that include physiological events, such as food intake and sleep, and to neuronal/physical activation. For example, neuroblast migration is promoted by ghrelin, a hormone secreted by the intestine in response to food intake [29]. The neurotransmitter serotonin released from activated serotonergic neurons promotes neuroblast migration in the RMS and the path toward lesions in the neocortex [28]. Interestingly, serotonin secretion is periodically altered by circadian rhythms. Therefore, several extrinsic factors affect the inconstant migration behavior of neuroblasts over multiple timescales in vivo, which in turn may contribute to the efficient migration of neuroblasts in adult brain tissue.

The inconstant migration behavior can also occur in a cell-autonomous manner. The efficiency of wound healing is influenced by the circadian modulation of actin-dependent phenomena such as cell migration and adhesion [37]. The expression of adhesion molecules in endothelial cells peaks in the evening in mice, activating the migration of leukocytes from the blood to the organs [38]. The migration of neuroblasts involves the regulation of actin polymerization [21,34] and on the expression of adhesion molecules such as integrin and N-cadherin [11,39], suggesting intrinsic inconstancy in their migration behaviors. Because actin and adhesion molecules can affect the saltation amplitude and adhesion strength, respectively, the inconstant situations generated in our simulations are feasible in vivo. Therefore, the inconstancy of migration behavior of V-SVZ-derived neuroblasts can likely be produced intrinsically and extrinsically in the brain, which helps efficient collective migration through the field with many astrocytes.

## 4.4. Perspectives

Our mathematical model does not incorporate mechanisms for directional control in order to allow for a range of migration scenarios. This simplification is a limitation of this study. In other models of collective cell migration, such as the lateral line of zebrafish embryos, melanoma cells in mice, and neural crest cells in frog embryos, collective formation makes

each cell faster and more directed, which may involve a chemotactic gradient formed by the cells at the rear of the collective [40]. Neuroblasts generated in the adult V-SVZ often form elongated clusters, occasionally along blood vessels, during long-range migration toward the injury site [4,22,24,25]. Future models integrating such rear-to-front signaling mechanisms and scaffold interactions may provide deeper insights into the directed collective migration. Additionally, beyond Slit-Robo signaling, other molecular mechanisms, such as laminin-β1 integrin signaling and various Eph-ephrin signaling pathways, have been reported to mediate interactions between neuroblasts and the surrounding astrocytes [11,41]. Incorporating these signaling pathways may further enhance our understanding of the mechanisms underlying neuroblast migration and neuronal regeneration.

Although neuroblast migration in the adult brain has been observed in various animal species, including some primates, the evidence of neuroblast migration in the adult human brain remains limited [42,43]. However, postmortem brain studies have reported enhanced cell proliferation in the V-SVZ and the presence of newly generated neurons near the lesion sites [44,45]. Enhancing the migration of endogenous or transplanted neural progenitor cells to injured sites promoted functional recovery in mouse models of ischemic stroke and traumatic brain injury [8,19,20,46]. In addition, neural progenitor cell transplantation improved functional outcomes in Huntington's disease [47,48] and cognitive function in mouse models of Alzheimer's disease [49,50]. For clinical applications, neuronal regeneration in the large human brain requires newly generated neurons, whether endogenous or transplanted, in order to effectively migrate from the origin to cover the injured areas. The microenvironment varies depending on the disease and its progression. Reactive astrocytes, as modeled in this study, proliferate in conditions such as ischemic stroke, neurodegenerative diseases, and traumatic brain injury; however, tissue changes following injury differ according to disease type, stage, and severity [51,52]. By incorporating these environmental factors, our model could predict neuronal migration dynamics across various pathological conditions, thus offering insights into the distribution and settlement of transplanted cells. This may aid in optimizing transplantation sites and timing, as well as in selecting extracellular interventions to improve therapeutic outcomes.

In the field of engineering, we propose a swarm robotics design for migration tasks. Conventionally developed swarming robots are expected to migrate over flat fields without obstacles [53,54]. These can function in prepared stages, but not in unstructured, sometimes inaccessible fields in nature. A possible robotic application of the modeled neuroblast-astrocyte system may be adaptable to such a situation: a number of elongated robots released in nature form a chain-like collective via physical contact and some attractive force; compared with individual robots, collective ones exert a stronger force to push environmental obstacles aside and create a narrow pass; the following robots reuse the pass, and many of the robots eventually reach the desired destination, such as a disaster or unexplored site. A recent study on worm collective behavior proposed a similar concept, in which a flexibly deformable collective composed of elongated soft robots explores and exploits complex natural environments [33]. In such swarm robot systems, we can apply additional properties that promote the migration efficiency, such as moderate adhesion and inconstancy.

## Supporting information

**S1 Appendix. Detailed methods.**
(PDF)

**S1 Table. Simulation parameters.**
(PDF)

**S1 Fig. Heat maps of regression slope $\alpha$ of forward speed $v_i(t)$ for collectivity $n_i(t)$ under various parameter values.** The neuroblast-related parameter $k_P$ (top left) and the astrocyte-related parameters $\mu_A$ (top right), $\tau$ (middle left), $\rho_A$ (middle right), $d_{slit}$ (bottom left), and $H_A$ (bottom left) are tested. The values of each parameter are shown on the left side of the heat map, and the asterisk indicates the value used in the main simulations and when the other parameters were altered. The parenthesized numbers for $\rho_A$ represent the lowest number of neuroblasts that causes an astrocytic unit to target the minimum radius. The "constant" column indicates the conditions where neuroblasts have a constant adhesion strength, that is, $A_{act} = 0$. The "periodic" and "periodic′" columns indicate periodic-inconstant conditions with $A_{act} = 5$ and $A_{act} = 10$, respectively, both with $\omega_{sal}/\omega_{act} = 36$. The "random" column indicates random-inconstant conditions with $\sigma_A = 5$ and $u = 3$. The number of astrocytes showing reactive shrinkage is $N_A = 40$. A large positive value on a blue background indicates that a neuroblast in a collective tends to be faster than a solitary neuroblast; a large negative value on a red background indicates the opposite tendency. The bold magenta value in the inconstant condition indicates a slope value greater than that under the constant condition for a common value of each tested parameter, representing the collective-advantageous tendency of inconstancy.
(PDF)

**S2 Fig. Heat maps of regression slope $\alpha$ of forward speed $v_i(t)$ for collectivity $n_i(t)$ under smooth morphology conditions.** Top: rounded-rectangular neuroblasts with the same values of $r_S$, $r_T$, and $w$ (5 μm in the snapshot). Bottom: exact-circle astrocytes, each with one astrocytic unit ($r_{max} = 56$, $\rho_A = 5.6$, and $r_{min} = 28$ in the snapshot). Size-related values are listed on the left side of each heat map. Heat maps are shown as in S1 Fig. The number of astrocytes showing reactive shrinkage is $N_A = 40$.
(PDF)

**S3 Fig. Simulations with neuroblasts initially clustered.** The forward speed $v_i(t)$ and collectivity $n_i(t)$ over time are averaged over ten trials. Target length $A_{sal}(t)$ and forward speed: dashed lines, constant; solid lines, periodic-inconstant. Collectivity: cross, constant; triangle: periodic-inconstant. In the inconstant condition, the wavelength of the activity cycle is set to 12 h, that is, $A_{act} = 5$ and $\omega_{sal}/\omega_{act} = 36$. The yellow, green, and cyan colors of the neuroblasts in the snapshots indicate the minimum, average, and maximum states in $A_{sal}(t)$, respectively. The number of astrocytes showing reactive shrinkage is $N_A = 40$.
(PDF)

**S4 Fig. Heat maps of regression slope $\alpha$ of forward speed $v_i(t)$ for collectivity $n_i(t)$ under various inconstant conditions of saltation amplitude ($A_{sal}(t)$).** Left: the values at the top indicate the number of saltation cycles per activity cycle ($\omega_{sal}/\omega_{act}$), whereas those on the left indicate the activity amplitude ($A_{act}$). Right: values at the top indicate the number of saltation cycles that the neuroblast repeats with a value of saltation amplitude ($u$), whereas those on the left indicate the standard deviation of the random change in saltation amplitude ($\sigma_A$). The number of astrocytes showing reactive shrinkage is $N_A = 40$. The adhesion strength between neuroblasts is constant ($H_N = 2.22$ [nN]). A large positive value on a blue background indicates that a neuroblast in a collective tends to be faster than a solitary one.
(PDF)

**S5 Fig. Heat map of regression slope $\alpha$ of forward speed $v_i(t)$ for collectivity $n_i(t)$ under different constancy conditions of adhesion strength between neuroblasts.** Values on the left indicate the number of astrocytes. Upper rows labeled "reactive shrink" show conditions in which astrocytic units shrink depending on the number of nearby neuroblasts. Lower rows

labeled "mismatched shrink" show the conditions where apparently random astrocytic units shrink. The left column indicates the conditions where neuroblasts have a constant adhesion strength, that is, $A_H = 0$. The middle column indicates periodic-inconstant conditions with $A_H = 8$ and $\omega_{sal}/\omega_{act} = 36$. The right column indicates random-inconstant conditions with $\sigma_H = 4$ and $u = 3$. The saltation amplitude is constant in these conditions ($A_{sal}(t) = A_{mid} = 10$). A large positive value on a blue background indicates that a neuroblast in a collective tends to be faster than a solitary one, while a large negative value on a red background represents the opposite tendency.
(PDF)

**S1 Video. Migration of the single neuroblast shown in Fig 1a.** Timelapse images are taken every 4 min for 128 min, as shown at the upper right (h:min). Scale bar = 10 μm.
(MP4)

**S2 Video. Enlarged simulations corresponding to Fig 6.** The four situations (a–d) are shown in order. The number on the top indicates the time [min] in each trial.
(MP4)

**S3 Video. Overview simulations related to Figs 6 and 7.** Upper row: $N_A$, the number of astrocytes. Middle row: property of astrocyte shrinkage. Lower row: property of constancy in saltation amplitude $A_{sal}(t)$. The inconstant case is a periodic condition with $A_{act} = 5$ and $\omega_{sal}/\omega_{act} = 18$. The adhesion strength is constant ($H_N = 2.22$ [nN]) in all the simulations.
(MP4)

**S4 Video. Enlarged simulations corresponding to Fig 8.** The three situations (a–c) are shown in order. The number on the top indicates the time [min] in each trial.
(MP4)

**S5 Video. Overview simulations showing the entire field related to Figs 8 and 9.** Upper row: $H_N$, adhesion strength between neuroblasts. In all the conditions, the total number of astrocytes is $N_A = 40$ with reactive shrinkage, and the saltation amplitude of neuroblasts is constant.
(MP4)

**S1 File. Codes for simulations and evaluation.** The simulation codes in C++ are named "main.cpp," "Monitor.cpp," and "Monitor.hpp" with a "makefile" file. The evaluation code in R is named "evaluation.Rmd." Each file is readable as a text file when the extension is changed to ".txt" alternatively.
(ZIP)

## Acknowledgments

We thank Dr. Daiki Umetsu (The University of Osaka), Dr. Yuichiro Sueoka (The University of Osaka), Dr. Taishi Mikami (Kajima Corporation), and Dr. Hiraku Nishimori (Meiji University) for the fruitful discussions. We thank Dr. Qiang Lu (Beckman Research Institute of the City of Hope) for providing the Dcx-DsRed mice. We thank Dr. Kazunobu Sawamoto (Nagoya City University) for the technical support. We thank Dr. Masahiro Shimizu (Nagahama Institute of Bio-Science and Technology) for providing part of the simulation codes. We thank Editage (www.editage.jp) for the English language editing.

## Author contributions

**Conceptualization:** Naoko Kaneko, Takeshi Kano.

**Data curation:** Daiki Wakita.

**Formal analysis:** Daiki Wakita.

**Funding acquisition:** Yuriko Sobu, Naoko Kaneko, Takeshi Kano.

**Investigation:** Yuriko Sobu.

**Methodology:** Daiki Wakita.

**Project administration:** Takeshi Kano.

**Resources:** Yuriko Sobu.

**Software:** Daiki Wakita.

**Supervision:** Takeshi Kano.

**Validation:** Daiki Wakita.

**Visualization:** Daiki Wakita.

**Writing – original draft:** Daiki Wakita, Naoko Kaneko.

**Writing – review & editing:** Yuriko Sobu, Takeshi Kano.

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
