## [Decision Letter · Decision Letter 0]

5 Dec 2024

PCOMPBIOL-D-24-01548

A mathematical model suggests collectivity and inconstancy enhance the efficiency of neuronal migration in the adult brain

PLOS Computational Biology

Dear Dr. Kano,

Thank you for submitting your manuscript to PLOS Computational Biology. After careful consideration, we feel that it has merit but does not fully meet PLOS Computational Biology's publication criteria as it currently stands. Therefore, we invite you to submit a revised version of the manuscript that addresses the points raised during the review process.

Please submit your revised manuscript within 60 days Feb 04 2025 11:59PM. If you will need more time than this to complete your revisions, please reply to this message or contact the journal office at ploscompbiol@plos.org. Please include the following items when submitting your revised manuscript:

We look forward to receiving your revised manuscript.

Kind regards,

Stacey D. Finley, Ph.D.

Section Editor

PLOS Computational Biology

Stacey Finley

Section Editor

PLOS Computational Biology

Feilim Mac Gabhann

Editor-in-Chief

PLOS Computational Biology

Jason Papin

Editor-in-Chief

PLOS Computational Biology

**Additional Editor Comments:**

The reviewers appreciate the model and its potential applications. However, several issues should be addressed, related to biological relevance, parameter values, and data for model training.

**Journal Requirements:**

Please ensure that the CRediT author contributions listed for every co-author are completed accurately and in full.

At this stage, the following Authors/Authors require contributions: Daiki Wakita, Yuriko Sobu, Naoko Kaneko, and Takeshi Kano. Please ensure that the full contributions of each author are acknowledged in the "Add/Edit/Remove Authors" section of our submission form.

**Reviewers' comments:**

Reviewer's Responses to Questions

**Comments to the Authors:**

Reviewer #1: This manuscript entitled "A mathematical model suggests collectivity and inconstancy enhance the efficiency of neuronal migration in the adult brain" investigates computationally collective behaviors of migrating neuroblasts in a crowd of astrocyte, after proposing the simplified mathematical model of this neuroblast-astrocyte system. Their computational simulations firstly demonstrates that their model reproduces neuroblast collective behaviors with which they can migrate faster than solitary neuroblasts in the densely populated astrocytes, and after that, investigated several features of this collective behavior. Although the model proposed in this manuscript is simple or the computations may not be technically heavy, I still would like to value the significance of this study. Firstly, I acknowledge that the first major result of this manuscript -- the computational demonstration of the power of collectiveness of neuroblasts (or the motile elements) to efficiently move through the crowds of astrocytes (or obstacles) (Sect. 3.1) -- provides the solid progress in this field. While I think this result is intuitively reasonable, it does not mean it is trivial. Secondly, as one of the results around the collective neuroblast migration, the prediction of the importance of the inconstant neuroblast behavior for the effective collective migration (Sect. 3.3) is interesting. Thus, this manuscript could be a good candidate for publication. That being said, I would like to reserve the recommendation until the authors address the following points (at least, the major comments):

Major comments.

1. In their model, the cells are made of the connected circles. In this case, because the peripheries of the elements have convex and concave shape, the two cells contacting each other may have the friction in between them when they slide relatively. I think this is not the effect which the authors desired in the model and not something controllable. I cannot judge if this effect brings something significant in their results. So, I wonder if the authors could the authors evaluate if this effect is negligible to their results and comment it in the manuscript, or if they could improve the model to avoid such bumpy periphery.

2. Most of the mathematical explanations of their model is currently put only in the supplementary file, not in the main text (Section 2). However, actually, this is making the manuscript difficult to follow smoothly. Considering this is the journal for “computational” biology, the authors may try to reorganize the manuscript and move the essential equations into the main text. For example, at least, I would put the equations of motions of the cells [Eqs. (7), (8) and (11) in the supplementary text], time-dependent friction to make each neuroblast motile [Eq. (9) in the supplementary text] and the decomposition form of the cell-cell interaction [Eq. (12) in the supplementary text] into the main text. (Regarding Eq. (12) in the supplementary text, this decomposition form of the interaction itself, F= sum f_kl + sum g_kl + sum g’_kl, may be moved to the main text, but the definition of each interaction term, f_kl, g_kl, g’_kl, would be explained by only the words in the main text whereas the detailed math would be still kept in the supplementary text.) This is because, without having these equations, the readers who have mathematical/physical backgrounds may feel difficulties to even just imagine what kind of principles were used to model their dynamics.

3. The authors may expand Section 3.3 to discuss what is the cause of the efficient collective behavior by the inconstant saltatory amplitude. Actually, they discuss it a bit in the Discussion (Section 4.2) by the sentence “[…] transient large-amplitude movements contributed to deadlock breaking, which increased the efficiency of the forward movement of the large neuroblast clusters (Fig. 6d)”, but this is just by words and Fig. 6d is just a snapshot. The authors may need to provide more quantifications or new simulations to support this proposition.

Minor comments.

Page 5 line 79 --- I guess that phi_i may be a typo, and it actually should be phi_{sal,i}

Page 8 line 141-152 --- I could not get the logic flow in between these two sentences. Maybe the authors forgot to put the sentence like “Each of these aspects (i)-(iv) can be explained in more details as follows.” or the phrase/sentence like that? I might be wrongly understanding what they intended.

Page 8 line 158-163 “By contrast, […] located (Fig. 3c)” --- I could not understand these sentences. There might be some typos?

Abstract, Introduction, etc. --- Apparently, to introduce a motivation of this computational study, the authors are frequently emphasizing that the observations of this system are challenges so the mathematical simulation study is a good alternative. However, I do not feel this is efficient reasoning of this study because the mathematical model used in this manuscript is pretty simplified one (2D only, cells as connected circles, etc) and there are no guarantees whether this model can quantitatively reproduce the experimental observations. And, if they really would like to emphasize so, they may need more rigorous comparisons of their model and experimental quantifications in this manuscript. Please note that, nevertheless, I do not want to undermine the significance of the study by this mathematical model. I think this model study is indeed providing the interesting concept, as mentioned above. So, I rather would like to suggest the authors to update the introduction so that it does not sound as if they want to study this mathematical model as an alternative of actual experiments but to predict something new which we had not realized.

Reviewer #2: This paper presents a mathematical model simulating the behavior of neuroblast migration in the adult brain, focusing on their collective behavior and interaction with astrocytes. The authors developed a two-dimensional model incorporating saltatory movement of neuroblasts, astrocyte shrinkage responses, and cell-cell interactions. Their simulations suggest that collective migration is advantageous in dense astrocyte environments, and temporal variations in neuroblast behavior ("inconstancy") can enhance migration efficiency.

Strengths:

1. Novel contribution to understanding collective cell migration mechanisms, particularly highlighting the role of "inconstancy" using mathematical simulation

2. Well-structured model incorporating key biological features (saltatory movement, Slit-Robo signaling, cell adhesion)

3. Potential applications in both medical research and swarm robotics

Weakness:

1. Two-dimensional model may not fully capture the complexity of three-dimensional brain tissue

2. There is very limited discussion on model sensitivity.

3. The astrocyte response is very simplified. The author only used a linear response to approximate the astrocyte dynamics and only consider the shrinkage behavior. However, actual astrocytes have highly complex, star-shaped morphology, exhibiting highly dynamic process extensions and retractions. Multiple signaling pathways may be involved.

4. There is a overemphasis on mathematical parameters without broader discussion of potential biological variations in neuroblast populations and various biological processes.

Major concerns:

1. The model's reliance on mathematical parameters, which are hand-tuned rather than experimentally derived, limits its biological applicability. Additionally, the findings may not readily translate to human neurobiology, where neuroblast migration is limited compared to rodents. To address the concern, the author should declare how reasonable the parameter setting is and add additional sensitivity analysis.

2. The simulated results are intriguing but fall short of being compelling or inspiring from a biological perspective without stronger connections to experimental evidence. There is insufficient discussion of how the simulated outcomes align with or explain in vivo and in vitro observations. This disconnect weakens the biological relevance and interpretability of the study, the authors should detail in the results section how the simulated findings correspond to specific experimental data. For instance, they could compare the model’s predictions with observed patterns in neuroblast migration under similar conditions in existing studies or suggest how their predictions might guide future experiments.

3. While the paper considers a range of factors (e.g., adhesion strength, astrocyte density, inconstancy in neuroblast behavior), it is unclear if these are comprehensive or represent the full biological complexity influencing neuroblast migration. The omission of potentially significant biological interactions or environmental conditions raises concerns about the realism and applicability of the simulated results to actual biological systems. To address this, the authors should either a) substantiate that the selected factors are indeed the primary determinants of neuroblast migration or b) explicitly discuss the potential limitations of the factors considered and how unaccounted variables might influence the outcomes.

Minor concerns:

1. It may be better if there are more than 10 repetitions.

2. Some technical points are unclear, for example, a) Missing details on numerical integration methods and time-step selection b) Unclear handling of periodic boundary conditions, c)No specification of computational requirements (runtime, memory usage)

3. Some figure presentiation could be refined: for example, a)Inconsistent scale bars and time stamps (especially in Figs 1, 6, and 8), b) Missing statistical significance indicators in regression analyses (Fig 7)

4. More discussion could be made on implications for neurodegenerative disease therapies and linking predictions to existing biological knowledge.

Reviewer #3: General comments:

In this manuscript, Wakita et al. presents a computational framework to understand how the neuronal migration process in the adult brain. They explore and propose a possible model by which neuroblasts achieve efficient migration through collective behavior and temporal variability. Using in silico modeling and experimental observations, the study highlights key mechanisms underlying neuroblast migration, including the role of astrocytic interaction, neuroblast collectivity, and behavioral inconsistency. The text and figures are relatively easy and clear to follow. However, while the idea is interesting, I did not see biological interpretation of the results discussed or implications of the work inferred; a major concern is how effectively a mathematical model can simulate neuronal migration from a biological perspective.

Major issues:

1. Reliability of the Mathematical Model: The foundation of any mathematical model predicting biological effects depends significantly on the robustness and reliability of the simulation. This, in turn, relies heavily on the integration of a substantial amount of biological data (e.g., images from this manuscript). However, the biological data presented here seems quite limited. Would it be possible to utilize published datasets from the literature as input to test the fidelity of this model? Additionally, could this model be tested using data on radial migration during embryonic development before applying it to the adult brain? Although the authors have emphasized the model's application to the adult brain, the extensive data available on embryonic development could serve as a valuable standard for testing.

2. in vivo vs. in vitro Data: The manuscript primarily relies on in vitro culture and brain slice experiments as input for computational simulations. While the authors highlight the similarities between in vivo and in vitro systems, these approaches are less representative of the in vivo environment, particularly for classical neuronal migration pathways, such as radial migration (cortical development) and tangential migration (adult neurogenesis), which are traditionally studied in vivo. Could publicly available in vivo datasets be incorporated to train the mathematical models? This would allow for further validation using the in vitro data, rather than the reverse approach.

3. Incorporating Biological Factors into the Model: The observations regarding neuroblast migration—such as the cellular and molecular factors involved—are indeed interesting. How do the authors plan to incorporate these biological factors comprehensively into the proposed mathematical model? Once established, could this model be used to predict biological phenotypes and provide valuable guidance for mechanistic studies in disease contexts?

Minor Comments:

1. Typographical Error in S1 Appendix Ic: There appears to be an inconsistency in the experimental description. The text states, "Using old male Gfap-EGFP mice aged 9–12 weeks, ischemic stroke was induced by the middle cerebral artery occlusion and brain sections were prepared at postnatal day 18 as described previously." This seems contradictory, as it would not be possible to prepare brain sections at postnatal day 18 in mice aged 9–12 weeks. This may be a copy-paste error. The description should likely read: "Using old male Gfap-EGFP mice aged 9–12 weeks, ischemic stroke was induced by middle cerebral artery occlusion, and brain sections were prepared at post-surgery day 18 as described previously."

**Have the authors made all data and (if applicable) computational code underlying the findings in their manuscript fully available?**

Reviewer #1: Yes

Reviewer #2: Yes

Reviewer #3: Yes

PLOS authors have the option to publish the peer review history of their article (what does this mean?). If published, this will include your full peer review and any attached files.

Reviewer #1: No

Reviewer #2: No

Reviewer #3: No

**Figure resubmission:**
---

## [Decision Letter · Decision Letter 1]

20 Apr 2025

PCOMPBIOL-D-24-01548R1

A mathematical model suggests collectivity and inconstancy enhance the efficiency of neuronal migration in the adult brain

PLOS Computational Biology

Dear Dr. Kano,

Thank you for submitting your manuscript to PLOS Computational Biology. After careful consideration, we feel that it has merit but does not fully meet PLOS Computational Biology's publication criteria as it currently stands. Therefore, we invite you to submit a revised version of the manuscript that addresses the points raised during the review process.

Please submit your revised manuscript within 30 days Jun 20 2025 11:59PM. If you will need more time than this to complete your revisions, please reply to this message or contact the journal office at ploscompbiol@plos.org. Please include the following items when submitting your revised manuscript:

We look forward to receiving your revised manuscript.

Kind regards,

Stacey D. Finley, Ph.D.

Section Editor

PLOS Computational Biology

**Journal Requirements:**

1) Please confirm whether your study includes live participants. If so, please insert an Ethics Statement at the beginning of your Methods section, under a subheading 'Ethics Statement'. It must include:i) The full name(s) of the Institutional Review Board(s) or Ethics Committee(s)ii) The approval number(s), or a statement that approval was granted by the named board(s). 2) Please include the affiliation of Takeshi Kano in the online submission form. 

**Reviewers' comments:**

Reviewer's Responses to Questions

Reviewer #1: In the revised version, the authors addressed all my comments in detail. I found the manuscript has been very much improved.

Reviewer #2: Overall, the authors have substantially improved the manuscript by addressing many of my concerns, particularly regarding parameter sensitivity analysis, model robustness, and potential clinical applications. The addition of supplementary simulations and videos enhances the presentation of their findings. Most of my previous concerns have been carefully addressed. I only have several extra comments that the author may take to make the manuscript more readable:

1. For the previous major concern 1, while the authors acknowledge hand-tuning parameters, they provide insufficient justification for why their selected parameter values lead to optimal biological behavior. I would recommend adding quantitative criteria for parameter selection and comparative evidence showing why these specific values best reproduce biological observations. Or, if the quantitative criteria experiment is difficult to perform, a more systematically description should be made in the main text on how the author actually "check visualized simulations agreed with biological observations" and why so.

2. In terms of writing, I think the author may consider adding some of the response to my previous major concern 3 (about why the certain important factors are selected) to the main text, before the introduction of mathematical formulation,

3. Just curious, is there any other molecular signals beyond Slit-Robo might be involved in the process? If there is, shouldn't they be included in the discussion part?

Reviewer #3: The authors' revisions have addressed my concerns.

**Have the authors made all data and (if applicable) computational code underlying the findings in their manuscript fully available?**

Reviewer #1: Yes

Reviewer #2: Yes

Reviewer #3: Yes

PLOS authors have the option to publish the peer review history of their article (what does this mean?). If published, this will include your full peer review and any attached files.

Reviewer #1: No

Reviewer #2: No

Reviewer #3: No

**Figure resubmission:**
---

## [Editor Report · Decision Letter 2]

2 May 2025

Dear Dr. Kano,

We are pleased to inform you that your manuscript 'A mathematical model suggests collectivity and inconstancy enhance the efficiency of neuronal migration in the adult brain' has been provisionally accepted for publication in PLOS Computational Biology.

Best regards,

Stacey D. Finley, Ph.D.

Section Editor

PLOS Computational Biology

---

## [Editor Report · Acceptance letter]

PCOMPBIOL-D-24-01548R2

A mathematical model suggests collectivity and inconstancy enhance the efficiency of neuronal migration in the adult brain

Dear Dr Kano,

I am pleased to inform you that your manuscript has been formally accepted for publication in PLOS Computational Biology. Your manuscript is now with our production department and you will be notified of the publication date in due course.

With kind regards,

Lilla Horvath
